# Analyzing 23 years of warm-season derechos in France: a climatology and investigation of synoptic and environmental changes

Lucas Fery[1,2,*] and Davide Faranda[1,3,4,*]

[1]Laboratoire des Sciences du Climat et de l'Environnement, UMR 8212 CEA-CNRS-UVSQ, Université Paris-Saclay, IPSL, 91191 Gif-sur-Yvette, France
[2]SPEC, CEA, CNRS, Université Paris-Saclay, F-91191 CEA Saclay, Gif-sur-Yvette, France
[3]London Mathematical Laboratory, 8 Margravine Gardens London, W6 8RH, UK
[4]Laboratoire de Météorologie Dynamique/IPSL, École Normale Supérieure, PSL Research University, Sorbonne Université, École Polytechnique, IP Paris, CNRS, Paris, 75005, France
[*]These authors contributed equally to this work.

**Abstract.** Derechos are severe convective storms known for producing widespread damaging winds. While less frequent than in the United States of America (USA), derechos also occur in Europe. The notable European event on 18 August 2022 exhibited gusts exceeding $200 \ \mathrm{km \, h^{-1}}$, spanning 1500 km in 12 hours. This study presents a first climatology of warm-season derechos in France, identifying thirty-eight (38) events between 2000 and 2022. Typically associated with a southwesterly mid-level circulation, warm-season derechos in France generally initiate in the afternoon and exhibit peak activity in July, with comparable frequencies in June and August. Predominantly impacting the northeast of France, these events exhibit a maximum observed frequency of 0.65 events per year, on average, within a 200 km by 200 km square region. These characteristics are similar to those observed in Germany, with notable differences seen in the USA, where frequencies can attain significantly higher values. The study also examines synoptic and environmental changes linked with analogues of the 500 hPa geopotential height patterns associated with past warm-season derechos, comparing analogues from a relatively distant past (1950–1980) with a recent period (1992–2022). For most events, a notable increase in convective available potential energy (CAPE) is observed, aligning with trends identified in previous studies for southern Europe. However, no consistent change in 0–6 km vertical wind shear is observed in the recent period. These environmental shifts align with higher near-surface temperatures, altered mid-level atmospheric flow patterns, and often, increased rainfall. The role of anthropogenic climate change in these changes remains uncertain, given potential influences of natural variability factors such as the El Niño Southern Oscillation (ENSO) or the Atlantic Multidecadal Oscillation (AMO).

## 1 Introduction

During the night of August 17 to 18, 2022, a mesoscale convective system (MCS) originated over the northern Balearic Islands, initially forming as a line of thunderstorms that gradually curved into a bow echo. This system swiftly propagated to the northeast, impacting Corsica in the early morning before reaching central and northern Italy, Slovenia, Austria, and the Czech Republic within a span of 12 hours. Notably, the convective system was sustained by the exceptionally warm

sea-surface temperatures (SST) of the Mediterranean Sea, leading to intense downbursts and surface wind gusts reaching up to 225 $\mathrm{km\,h^{-1}}$ in Corsica. The severity and extensive destruction caused by this extraordinary storm caught the general public by surprise. The event, resulting in 12 casualties, over 100 injuries, and disruptions to electric power lines (Wikipedia), prompted immediate inquiries into its uniqueness, comparisons with similar past events, and considerations of the potential role of anthropogenic climate change in contributing to such occurrences. Specifically, the warming of the Mediterranean Sea, characterized by anomalies exceeding +3°C during the summer compared to seasonal values for the period 1990–2020, has been identified as a significant contributor to the development and intensity of the derecho-producing MCS on 18 August 2022 (González-Alemán et al., 2023). This warming likely enhanced convective instability by supplying substantial moisture and heat to the lower levels of the atmosphere. This storm aligns with the definition of a "derecho" as described by Hinrichs (1888), a term denoting convective storm episodes marked by prolonged and widespread occurrences of damaging downbursts. More specifically, a derecho is defined as "any family of downburst clusters generated by an extratropical mesoscale convective system," in accordance with Johns and Hirt (1987). The associated radar signatures generally exhibit predominantly linear characteristics, often incorporating bow echoe(s) indicative of regions with the highest wind speeds (Fujita, 1978). Derechos are commonly classified into two types: "serial" and "progressive" (Johns and Hirt, 1987; Corfidi et al., 2016; Squitieri et al., 2023a). A "progressive" derecho is characterized by a rapidly propagating MCS featuring a long-lived bow echo pattern, nearly perpendicular to the mean wind direction. Typically occurring in the warm season (May–August), this type of derecho propagates faster than the mean wind and is associated with high convective instability. Key features include a rear-inflow jet and mesoscale vortices. In contrast, a "serial" derecho typically showcases an extensive squall line with a Line Echo Wave Pattern (LEWP) (Nolen, 1959) embedded within a cold front. This type tends to occur in the cold season (September–April) in an environment characterized by strong forcing and low instability. Convective systems of this type generally propagate more slowly than "progressive" derechos.

Various criteria have been employed to delineate a sequence of downburst clusters as a derecho, using observational data such as wind gust reports and weather radar data. For instance, Johns and Hirt (1987) proposed the following criteria: i) a concentrated area of convective wind gust with a speed greater than or equal to 26 $\mathrm{m\,s^{-1}}$ or—when wind-speed measurements are not available—the presence of damage following downbursts. The major axis length must be of at least 400 km; ii) the convective gusts must have an identifiable spatio-temporal progression, iii) at least 3 gusts greater than or equal to 33 $\mathrm{m\,s^{-1}}$ must be measured or assessed on the basis of damage within the area covered by the episode and these reports must be separated by at least 64 km from each other; iv) there must be no more than 3 hours between two consecutive severe wind gust reports; v) the associated convective system must have temporal and spatial continuity in surface pressure or wind field; vi) all the wind gust reports must emanate from the same MCS based on radar data.

Nevertheless, subsequent studies have frequently adjusted or relaxed the criteria outlined above, particularly the most restrictive criterion (iii). It was argued by Bentley and Mote (1998) that there is no explicit reference to wind threshold criteria in the common definition of derechos as a family of downburst clusters by Fujita and Wakimoto (1981). Notably, Coniglio and Stensrud (2004), Bentley and Mote (1998), and Gatzen et al. (2020) have opted not to retain this criterion. Instead, they assigned intensity to events based on the number of wind gust reports at different thresholds of wind speed.

Corfidi et al. (2016) proposed a more stringent definition of derechos, focusing exclusively on the most severe, long-lived "progressive" or warm-season derechos. This distinction arises from the different dynamics and environments associated with "progressive" and "serial" derechos. Serial derechos are associated with "external" synoptic-scale forcing for ascent, while progressive derechos are characterized by more "internally driven" dynamics. The persistence of progressive derechos is attributed to the presence of mesoscale features such as a rear-inflow jet, a well-defined surface cold pool, and the swift downstream propagation of new updrafts along the gust front of the cold pool (Corfidi, 2003; Schumacher and Rasmussen, 2020). The refined definition proposed by Corfidi et al. (2016) stipulates that the damage swath "must be nearly continuous, at least 100 km wide along most of its extent, and 650 km long". Additionally, this definition requires clear evidence of radar features, including bow echoes, mesoscale vortices, and rear-inflow jets. For a comprehensive overview of research on derechos, including the varied criteria used in past studies, the reader is referred to the recent review by Squitieri et al. (2023a, b).

Derechos are primarily documented phenomena in the USA, particularly in the Midwest and Southern Plains, as noted by various studies (Hinrichs, 1888; Johns and Hirt, 1985, 1987; Bentley and Mote, 1998; Evans and Doswell, 2001; Coniglio and Stensrud, 2004; Ashley and Mote, 2005; Guastini and Bosart, 2016). In contrast, the scientific exploration of derechos in Europe is more recent, with events being officially recognized and reported as such since the 2000s (Gatzen, 2004; Punkka et al., 2006; López, 2007; Púčik et al., 2011; Hamid, 2012; Celiński-Mysław and Matuszko, 2014; Mathias et al., 2019). Despite this, comprehensive national climatologies are scarce in Europe (Gatzen et al., 2020; Celiński-Mysław et al., 2020), and research on derechos in France is notably limited. While individual cases have been reported in scientific articles (López, 2007; Hamid, 2012; Gatzen et al., 2020) and in weather reports from Keraunos, the French observatory of severe convective storms (website: https://www.keraunos.org/), there is currently no previous systematic study of derecho climatology in the country. Recognizing this gap in research motivates our current study, where we aim to provide a first analysis of derecho occurrences in France and contribute to the broader understanding of these phenomena in the European context.

Unraveling the nuanced impact of anthropogenic climate change on severe convective events poses another formidable challenge (National Academies of Sciences, Engineering and Medicine, 2016), particularly when it involves discerning trends in convective hazards like convective winds (Kunkel et al., 2013). Because of their complexity involving a large range of time and spatial scales and various physical phenomena such as wind gust, precipitation and lightning, MCS are not well resolved in global climate models while they can be better simulated in convection-permitting, regional climate models although at a high computational cost (Meredith et al., 2015; Gensini and Mote, 2015; Gensini et al., 2023; Coppola et al., 2020; Ban et al., 2021; Stocchi et al., 2022). It is therefore difficult to find clear climate change statements about severe convective storms, including derechos. Due to these modelling difficulties, even the IPCC reports do not contain many strong statements about the influence of anthropogenic climate change on severe convective events. Indeed, in the AR6 report (Intergovernmental Panel On Climate Change (IPCC), 2023), the authors find that there is high confidence that "a warmer climate intensifies very wet and very dry weather events and seasons, but the location and frequency of these events depend on projected changes in regional atmospheric circulation". Particularly for Europe, there is moderate confidence that at 1.5°C of warming, "heavy precipitation and associated flooding are projected to intensify and be more frequent", and low confidence that "large-scale conditions conducive to severe convection will tend to increase in the future climate".

Numerous studies have delved into exploring potential changes in the frequency and intensity of MCS (Schumacher and Rasmussen, 2020). Additionally, investigations into environmental factors influencing convection, such as CAPE and vertical wind shear (Púčik et al., 2017; Taszarek et al., 2021a, b; Glazer et al., 2021), or convective hazards (Battaglioli et al., 2023; Gensini and Mote, 2015; Gensini et al., 2023; Pichelli et al., 2021), have been conducted in the context of global warming. Although the findings exhibit significant regional variations, there is a broad consensus indicating an increase in rainfall rate and volume associated with MCS under global warming conditions (Schumacher and Rasmussen, 2020). Within the northern midlatitudes, studies indicate a rise in CAPE over southern Europe and the northern Great Plains of the USA, accompanied by a marginal decrease in 0–6 km wind shear over southern Europe and a slight increase in the Great Plains (Taszarek et al., 2021a, b). However, these regions also witness an increase in Convective Inhibition (CIN) and a decline in relative humidity (Taszarek et al., 2021a; Pilguj et al., 2022), leading to an overall reduction in the fraction of environments conducive to storm initiation. This complex interplay makes definitive statements about the frequency of severe thunderstorms challenging, as highlighted by previous research (Kunkel et al., 2013; Taszarek et al., 2021a; Pilguj et al., 2022). The absence of unequivocal conclusions regarding the intensity and frequency of severe convective storms, including derechos, under anthropogenic climate change motivates the analysis presented in this study.

This paper aims to create a first climatology of warm-season derechos in France, analyzing their characteristics in comparison with other countries as its primary objective. The secondary goal is to detect potential alterations in synoptic conditions and environmental convective parameters, including CAPE and vertical wind shear, linked to past warm-season derecho-producing MCS in France. The study also seeks to evaluate the contributions of climate change and natural variability to these observed changes. In Sect. 2, we delve into the methodological aspects of our work and introduce the datasets employed. Specifically, Sect. 2.1 outlines the methodology and observational datasets used for detecting past derecho events over France, along with associated limitations. Additionally, Sect. 2.2 introduces the methodology for detecting and attributing changes in synoptic conditions, accompanied by the presentation of the reanalysis and observational datasets. Moving on to Sect. 3, we present the outcomes related to the detected derechos, their frequency, intensity, and geographical distribution in comparison with established climatologies in Germany or the USA (Sect. 3.1). Furthermore, we discuss the detailed attribution for the case of the 18 August 2022 derecho (Sect. 3.2) and provide an overall attribution for all events with interpretations (Sect. 3.3). Finally, conclusions and future perspectives are presented in Sect. 4.

## 2  Data and Methods

### 2.1  Derecho detection

Following the methodology employed by Gatzen et al. (2020) in establishing a derecho climatology for Germany from 1997 to 2014, we utilize daily weather station data from Météo-France (automatic stations of type 0 and 1). Our initial step involves selecting warm-season days (May, June, July, August, September) when more than 5 stations report a severe daily wind gust, defined as a measured wind gust speed exceeding $25\,\mathrm{m\,s^{-1}}$. Following this, we eliminate days without a concentrated area of wind gust reports by applying the criterion that the area must cover at least 400 km along its major axis. Additionally,

we stipulate that wind gust reports should be within 200 km of each other, and that there should be no more than a 3-hour interval between successive reports, aligning with the methodology employed by Coniglio and Stensrud (2004). To circumvent the limited geographical coverage of wind reports to France, we use severe wind gust reports from three additional sources: (i) the Integrated Surface Database (ISD) (Smith et al., 2011) from the National Oceanic and Atmospheric Administration (NOAA) (accessible from https://www.ncei.noaa.gov/products/land-based-station/integrated-surface-database), composed of

worldwide surface weather observations including synoptic observations; (ii) weather station data from the German Weather Service (DWD), accessible at https://cdc.dwd.de/portal) and (iii) the European Severe Weather Database (ESWD) (accessible at https://eswd.eu/) created by the European Severe Storm Laboratory (ESSL) (Dotzek et al., 2009) whenever the associated MCS track extends into other countries. The ESWD provides detailed and quality-controlled reports from severe convective events in Europe including severe wind gusts, heavy rain, hail, tornadoes, and damaging lightning from a variety of sources.

For our study, we only retained reports at minimum level quality QC1 ("report confirmed by reliable source"). These reports originate from various channels, including weather stations providing specific wind gust speeds and wind gust damage reports, such as those involving fallen trees. However, the latter type of report lacks precise wind gust speed information, impeding the accurate estimation of derecho intensity. In cases where reports are insufficient, we also consider days on which derechos have been reported over France by Gatzen et al. (2020).

Then we use the Python FLEXible object TRacKeR (PyFLEXTRKR) algorithm developed by Feng et al. (2023a) to systematically detect and track potential associated MCS for each previously selected day. This algorithm has notably been used to build a global MCS database (Feng et al., 2021) using satellite imagery data, namely brightness temperature and precipitation. It has also been used to track MCS in convection-permitting simulations (Feng et al., 2023b) or convective cells from radar data (Feng et al., 2022). To detect and track MCS, the algorithm uses brightness temperature thresholds to identify cold cloud

systems ($T_b < 241$ K) with an additional size constraint of an area greater than $4 \times 10^4$ km$^2$. Precipitation data is used in addition to enable a more robust identification of MCS by requiring that an intense precipitation feature (criteria includes rain rate greater than 3 $\mathrm{mm\,h^{-1}}$ and major axis length greater than 100 km) is embedded within the cold cloud system. These criteria must be met for at least 4 hours to define a cold cloud system as an MCS. We specifically use the Global Precipitation Measurement (GPM) Integrated Multi-satellitE Retrievals (IMERG) V06 precipitation database (Huffman et al., 2019) and the

NOAA NCEP/CPC Global Merged IR (MERGIR) brightness temperature ($T_b$) database (Janowiak et al., 2017), as in Feng et al. (2021). Both datasets are available from the year 2000 and cover the area between 60°S and 60°N at a time resolution of 30 minutes. The MERGIR database has a finer resolution (4km) than IMERG (10 km/0.1°), so we regrid $T_b$ data to a resolution of 0.1° using xESMF python package (Zhuang et al., 2023) prior to applying the tracking algorithm. If PyFLEXTRKR detects an MCS for a specific selected date, we conduct a visual comparison, aligning the detected MCS structure, as illustrated on

brightness temperature and precipitation rate maps, with the severe wind gust reports in both time and space. For illustration, a snapshot of $T_b$ and precipitation rate, highlighting the contour of the detected MCS using PyFLEXTRKR, is provided in the supplementary material (Fig. S1). Subsequently, we retain only those days when the reports exhibit a discernible match with an MCS along a minimum distance of 400 km. Additionally, we assign to each detected derecho an intensity following the criteria set by Gatzen et al. (2020) and Coniglio and Stensrud (2004). Specifically, an event is deemed high intensity if there

are at least 3 reports with wind speed $\geq 38$ ms$^{-1}$, moderate intensity if the last condition is not met and at least 3 reports show wind speed $\geq 33$ ms$^{-1}$, and low intensity if the event does not meet the previous criteria. Similar to the methodology of Gatzen et al. (2020), the trajectories, path lengths and duration of all derechos are determined by analyzing respectively the direction and length of the geodesic connecting the geographical locations of the first and last wind gust reports, and the time elapsed between these two reports. While this method offers a coarse estimate of the actual path of the MCS, it does not enable

a precise evaluation of the direction of MCS propagation. In particular it does not consider potential curvature in the track. Limitations in the available severe wind gust reports also constrain the accuracy of duration and path length estimations for derechos, introducing potential biases due to underreporting in certain regions.

In contrast to the methodologies employed by Coniglio and Stensrud (2004) and Gatzen et al. (2020), who utilized radar data for convective system identification, we opted for satellite data due to its greater accessibility (from e.g. https://disc

.gsfc.nasa.gov/), and its large spatial coverage. Radar data, commonly managed and hosted by national weather services, typically restrict coverage to national geographical domains, and their accessibility is often limited. Radar data would be crucial for studying the specific shape of MCS (e.g., detecting bow echoes), ensuring wind gusts originate from the same convective system, and precisely aligning the timestamps of wind gust reports with radar echo positions. Consequently, our methodology may not effectively distinguish between a swath of wind gusts produced by a well-organized bow echo and a

more disorganized convective cluster. Given our focus on warm-season derechos, the use of radar data becomes less critical compared to cold-season derechos, where distinguishing damaging winds related to downbursts from concurrent synoptic-scale winds is challenging (van den Broeke et al., 2005; Gatzen et al., 2020). Coniglio and Stensrud (2004) demonstrated that the MCS associated with warm-season derechos can be identified quite well even without radar data. Notably, as there has been some debate regarding whether the specific structure of the convective system producing a swath of severe wind gusts should

be considered in defining a derecho-producing MCS (Bentley and Mote, 1998; Johns and Evans, 2000; Bentley et al., 2000; Coniglio and Stensrud, 2004; Corfidi et al., 2016), we assert that our approach is reasonable for establishing a first climatology of these severe convective windstorms in France.

## 2.2 Detection of changes in synoptic patterns

To understand how anthropogenic climate change may have influenced the synoptic patterns associated with severe convective

events like derechos, we consider analogues of patterns of atmospheric circulation. While a direct one-to-one correspondence between synoptic patterns and derecho occurrences may not exist, these patterns commonly represent recurring large-scale conditions conducive to the development of severe convective events (Bentley et al., 2000; van Delden, 2001; Coniglio et al., 2004; Lewis and Gray, 2010; Markowski and Richardson, 2010; Yang et al., 2017; Mohr et al., 2019; Piper et al., 2019; Schumacher and Rasmussen, 2020). We use the geopotential height field at 500 hPa (Z500) as a proxy of large-scale atmospheric

flow, in a similar way as Burke and Schultz (2004), Coniglio et al. (2004) and Gatzen et al. (2020). To isolate the dynamical changes in the atmospheric circulation patterns, the average thermodynamic contribution of global warming is removed by subtracting its spatial mean from each daily geopotential height field. Indeed, this mean value exhibits a trend associated with anthropogenic climate change (Christidis and Stott, 2015). This approach allows us to focus specifically on changes in the

Z500 gradient, a parameter intricately tied to mid-level atmospheric flow. Although synoptic patterns predominantly influence the environmental factors driving diverse extreme events, such as the formation of derecho-producing MCS, it is crucial to acknowledge the significant role of sub-synoptic scale environments in convective storm development. This becomes particularly pertinent in the context of convective initiation and the subsequent release of CAPE (Markowski and Richardson, 2010).

More specifically, for each event, we look for Z500 analogues in both a relative distant past period (1950–1980) and a more recent past period (1992–2022). Our assumption in dividing the historical past in two periods is that the most distant period serves as a hypothetical world where the Earth's climate was only weakly affected by greehouse gas emissions, and that 31 years is a sufficient period to account for natural variability in atmospheric motions. This time period is also recommended by the WMO for the computation of climate normals (Arguez and Vose, 2011). Nevertheless, it is crucial to account for long-term natural variability, as induced by phenomena like the Atlantic Multidecadal Oscillation (AMO) or the El Niño-Southern Oscillation (ENSO). If we can exclude a direct impact from such low-frequency variability by examining the indices related to these phenomena alongside the analogues between the two periods under investigation, any changes in the analogues can be attributed to the signal of climate change. While conventional statistical techniques, rooted in extreme value theory, focus on analyzing univariate meteorological variables without tracing them back to the underlying atmospheric processes, our approach ensures that comparisons of variable maps are conditioned on the associated atmospheric circulation. Additionally, the method enables the identification of unprecedented weather events resulting from previously unobserved atmospheric circulations, posing a statistical challenge in attributing the event's likelihood to climate change. The attribution methodology described in Faranda et al. (2022) has already been applied and validated for sea-level pressure maps linked to various extreme events in 2021. In this study, we extend its application for the first time to synoptic patterns related to severe convective events, with a particular focus on derechos.

In our examination, we analyze changes in temperature, precipitation, and wind speed, along with proxies and environmental parameters commonly utilized as predictors for convection associated with these analogues within the historical period. Specifically, we focus on the most unstable Convective Available Potential Energy (CAPE) and 0–6 km vertical wind shear, calculated as the wind vector difference between 500 hPa and 10 m, referred to as deep-layer shear (DLS). CAPE serves as a proxy for buoyant instability, intricately linked to the intensity of convective updrafts (Holton and Hakim, 2013), while vertical wind shear fosters the organization of convection (Markowski and Richardson, 2010; Schumacher and Rasmussen, 2020). Numerous studies have demonstrated an augmented probability of severe convection with increasing levels of instability and vertical wind shear, frequently employing CAPE and DLS as metrics (Brooks et al., 2003; Trapp et al., 2007; Brooks, 2013; Púčik et al., 2015; Taszarek et al., 2020b; Battaglioli et al., 2023). We use data from ERA5 (Hersbach et al., 2020), the latest climate reanalysis produced by the European Centre for Medium-Range Weather Forecasts (ECMWF) as part of the implementation of the EU-funded Copernicus Climate Change Service (C3S). It provides hourly data on atmospheric, land surface and sea state parameters from 1950 to the present. The ERA5 data are available from the C3S Climate Data Store on regular latitude-longitude grids at a horizontal resolution of $0.25° \times 0.25°$. We opted for ERA5 data in this study primarily due to the dataset's remarkable consistency over an extensive time span (73 years), facilitating the detection of changes in large-scale dynamics. The global nature of ERA5 also mitigates issues related to combining datasets from diverse national weather services,

ensuring a uniform spatial and temporal coverage. Nevertheless, it is important to acknowledge certain caveats arising from the long-term advancements in observation instruments, including satellites. These advancements may impact the uniformity of the quality of the dataset and have the potential to introduce spurious trends (Thorne and Vose, 2010; Hersbach et al., 2020). In addition, parameterizations can bring errors in the estimation of convective parameters such as CAPE and modeled precipitation and wind gusts (Taszarek et al., 2021c). Despite these considerations, ERA5 continues to be regarded as one of the best reanalyses currently available.

For each period, we examine all daily averaged Z500 maps and select the best 37 analogues, i.e. the maps minimizing the Euclidean distance to the event map itself. The number of 37 corresponds approximately to the smallest 8 ‰ Euclidean distances in each subset of our data. We tested the extraction of 25 to 50 analogous maps, without finding qualitatively important differences in our results. For the factual period, as is customary in attribution studies, the date of the event considered is discarded. In addition, we prohibit the search for analogues within a one-week window centered on the date of the event. We also restrict the search for analogues to the season in which each event occurs (in this case the warm season comprising May, June, July, August and September). This allows us to identify possible changes in seasonality—defined as the relative frequency of analogues occurrence per calendar month—between the counterfactual and factual periods, while avoiding confounding the different physical processes that may contribute to a given class of extreme events during warm and cold seasons. We then examine composites of the analogues in each period for daily averaged 2 meter temperature and wind speed, daily cumulative precipitation, daily maximum CAPE and DLS all computed from ERA5 hourly fields. For comparison, we also consider daily mean 2 meter temperature and cumulative rainfall from the E-OBS observational dataset v27.0 (Cornes et al., 2018) (available from https://www.ecad.eu/download/ensembles/download.php) which interpolates measurements from land weather stations across Europe on a regular grid at 0.1° resolution. To determine significant changes between the two periods, we adopt a bootstrap procedure which consists of pooling the dates from the two periods together, randomly extracting 37 dates from this pool 1000 times, creating the corresponding difference maps and marking as significant only grid point changes above two standard deviations of the bootstrap sample.

To account for the possible influence of low-frequency modes of natural variability in explaining the differences between the two periods, we also consider the possible roles of ENSO, the AMO, the Pacific Decadal Oscillation (PDO), the North Atlantic Oscillation (NAO), and the East Atlantic (EA) and Scandinavian (SCAND) North Atlantic patterns. Piper et al. (2019) found that convection-favoring environments are strongly influenced by teleconnection patterns such as NAO, EA and SCAND and SST in Europe. The NAO, the EA and to a lesser extent ENSO have also been found to have a significant role in modulating extreme precipitation events in Europe by Nobre et al. (2017). The role of AMO has been discussed, e.g., in Zampieri et al. (2017) who found an influence on pressure, precipitation and temperature patterns. Wei et al. (2021) found an influence of PDO on northwestern Europe extreme rainfall. Similarly, Casanueva et al. (2014) found a significant role of SCAND in autumn and spring on extreme precipitation in Europe. In practice, we examine the association of the analogues with the aforementioned factors of natural variability (ENSO, AMO, PDO, NAO, EA, SCAND). We perform this analysis using monthly indices from NOAA/ERSSTv5 data and retrieved from the Royal Netherlands Meteorological Institute (KNMI) Climate Explorer (accessible at https://climexp.knmi.nl/selectindex.cgi). In particular, the ENSO index is computed in region 3.4 as defined by Huang

et al. (2017), and the AMO index is calculated as described in Trenberth and Shea (2006). To assess the possible association of these different indices on circulation changes between factual and counterfactual periods, we compare the distributions of each index for the analogues of the two periods and we evaluate any significant changes between factual and counterfactual distributions by performing a two-tailed Cramér-von Mises test (Anderson, 1962) at the 0.05 significance level. If the p-value is smaller than 0.05, the null hypothesis (H = 0) that both samples are from the same distribution is rejected, and the influence of internal variability cannot be excluded (H = 1). On the other hand, if the null hypothesis of equal distributions is not rejected, the observed changes in the analogues are attributed to anthropogenic climate change. All relevant figure panels display the p-value (pval) and the H-test results in the title.

We further investigate the seasonality of analogues within the warm season by quantifying the number of analogues in each month, aiming to identify potential shifts in circulation towards earlier or later months in the season. Such shifts could carry significant thermodynamic implications; for instance, if a circulation pattern associated with substantial positive temperature anomalies in early spring becomes more prevalent later in the season when average temperatures are considerably higher. To evaluate the significance of potential changes in analogue seasonality between the two periods, we apply the same statistical test that we use to compare the distributions of indices of natural variability. Additionally, we extend our analysis by computing the best 8 ‰ analogues for the entire Z500 dataset from 1950 to 2022 without segregating it into factual and counterfactual periods. Subsequently, we estimate a linear trend for this global quantile, recognizing that the total number of analogues in all decades for this specific quantile is 91. To evaluate the significance of these trends, we calculate the confidence interval using the Wald method (Stein and Wald, 1947).

Following Faranda et al. (2022), we define certain quantities that support our interpretation of analogue-based assignment. All these quantities can then be compared between the counterfactual and factual periods.

- **analogue quality Q**: Q is the average Euclidean distance of a given day from its 37 closest analogues. To evaluate the quality of the analogues, we compare Q for the day of each event to the distribution of the same metric computed for each of its analogues. If the value of Q for the day considered stands within the distribution of Q for its analogues, then the quality of the analogues is considered good. The event is not unprecedented and the attribution can be performed. However, if the value of Q for the event is greater that that of its analogues, the quality of the analogues is considered low, and the event is unprecedented and therefore not attributable.

- **Predictability Index** $D$. Using dynamical systems theory (Freitas et al., 2011, 2016; Lucarini et al., 2016), we can compute the local dimension D of each Z500 map (Faranda et al., 2017a, 2019). The local dimension is a proxy for the number of degrees of freedom of the field, meaning that the higher D, the more unpredictable the temporal evolution of the Z500 maps will be (Faranda et al., 2017b; Messori et al., 2017; Hochman et al., 2019). If the dimension D of the pattern associated with a derecho event analyzed is higher or lower than that of its analogues, then the extreme will be respectively less or more predictable than the closest dynamical situations identified in the data.

- **Persistence index** Θ: Another quantity derived from dynamical systems theory is the persistence Θ of a given configuration (Faranda et al., 2017a). Persistence provides an estimate of the number of days we are likely to encounter a map

that is an analogue of the one under consideration (Moloney et al., 2019). As with Q and D, we compute the two values of persistence for the extreme event in the factual and counterfactual world and the corresponding distributions of the persistence for the analogues.

## 3 Results

### 3.1 Detected derechos over France between 2000 and 2022

In total, we identify thirty-eight (38) warm-season derechos in France from 2000 to 2022. Table 1 presents a summary of these derechos, including their start date and time, path length, duration, intensity, and affected countries.

The trajectories of warm-season derechos in France, illustrated as straight arrows in Fig. 1a, predominantly follow a northeastern direction. Additionally, upon examining the daily averaged 500 hPa geopotential height patterns associated with the events (not shown), we predominantly observe southwesterly flows. This is consistent with the favored development of extratropical MCS ahead of a trough or a cut-off low (van Delden, 2001; Gatzen, 2004; Coniglio et al., 2004; Gatzen, 2013; Yang et al., 2017; Houze, 2018; Piper et al., 2019; Gatzen et al., 2020). Many Z500 patterns associated with warm-season derechos in France bear resemblance to the Spanish Plume (Morris, 1986; Holley et al., 2014), a typical configuration known for its association with severe convective weather in Northwestern Europe. Examining more precisely the map of tracks (Fig. 1a), we note a significant number of derechos that propagate from the southeastern and eastern regions of France, advancing towards the east-northeast, often reaching Switzerland and/or southern Germany. These derechos commonly traverse north of the Alps or the Jura Mountains, aligning with the majority of warm-season events identified by Gatzen et al. (2020). Another set of derechos originates in central and northern France, extending through Belgium, Luxembourg, and the Netherlands, with some instances affecting western and northern Germany, and a few persistent events extending as far as Denmark and Sweden. Additionally, certain derechos initiate in the southwest of France or northern Spain, propagating in various directions from north to east. The 2022 derecho impacting Corsica is distinctive, originating near the Balearic Islands in the Mediterranean Sea with no comparable event identified. The derecho of 17 August 2003 serves as the closest comparable event, albeit with a shorter path positioned further to the north. It originated in Spain and reached France after traversing the Gulf of Lion. It is possible that events similar to the 2022 derecho occurred but went undetected due to the limited number of weather stations over the sea. This particular derecho also affected multiple countries, suggesting the potential for identifying similar events by focusing on Mediterranean countries like Italy and Spain.

On average, we observe 1.65 derechos impacting France per warm season. However, regional variations are significant, as depicted in Fig. 1b, where we show the average number of warm-season derechos per year per $200\,\mathrm{km} \times 200\,\mathrm{km}$ square regions, following the approach of Gatzen et al. (2020) and Coniglio and Stensrud (2004). This involves counting the occurrences of associated severe wind gusts within each cell. Notably, the northeast of France exhibits the highest frequency of events, peaking at 0.65 derechos per year, while Brittany, the westernmost part of France, and Côte d'Azur, in the southeast, record no events. After closely examining the spatial distribution of observed derecho frequencies in Fig. 1b, we find a partial alignment with the European climatology of lightning in summer, as presented in Taszarek et al. (2020a) and with the density map of MCS

**Table 1.** List of warm-season derechos that affected France between 2000 and 2022. The countries are abbreviated with their ISO 3166-1 alpha-2 code.

| Number | Start date and time (UTC) | Path length (km) | Duration (h) | Intensity | Affected countries |
|---|---|---|---|---|---|
| 1 | 2 July 2000 13:55:00 | 560 | 7 | moderate | FR, BE, LU, DE |
| 2 | 6 July 2001 16:45:00 | 500 | 6 | moderate | FR, CH, DE |
| 3 | 15 August 2001 13:15:00 | 700 | 10 | low | FR, BE |
| 4 | 19 May 2003 12:45:00 | 960 | 8 | moderate | FR, CH, DE, CZ |
| 5 | 14 June 2003 05:17:00 | 1000 | 15 | moderate | FR, DE, AU |
| 6 | 15 July 2003 17:27:00 | 460 | 7 | moderate | FR |
| 7 | 17 August 2003 07:30:00 | 900 | 13 | moderate | ES, FR, CH |
| 8 | 28 August 2003 18:45:00 | 490 | 5 | moderate | FR, CH |
| 9 | 17 August 2004 14:42:00 | 450 | 6 | low | FR |
| 10 | 29 July 2005 14:27:00 | 740 | 10 | high | FR, CH, DE, CZ |
| 11 | 4 July 2006 19:33:00 | 410 | 4 | low | FR |
| 12 | 17 September 2007 15:59:00 | 420 | 5 | low | FR |
| 13 | 25 May 2009 21:42:00 | 990 | 11 | moderate | FR, BE, NL, DK |
| 14 | 12 July 2010 04:30:00 | 1260 | 13 | high | FR, BE, LU, NL, DE, DK |
| 15 | 14 July 2010 12:11:00 | 660 | 10 | moderate | FR, BE, LU, NL, DE |
| 16 | 22 June 2011 13:08:00 | 470 | 5 | moderate | FR, CH, DE |
| 17 | 7 June 2012 11:24:00 | 600 | 7 | low | FR, CH, DE |
| 18 | 20 June 2013 13:50:00 | 550 | 9 | low | CH, FR, DE |
| 19 | 26 July 2013 21:42:00 | 790 | 10 | moderate | FR, BE |
| 20 | 27 July 2013 16:23:00 | 620 | 7 | moderate | FR, BE, LU, NL, DE |
| 21 | 4 July 2014 13:49:00 | 510 | 7 | low | FR, CH, DE |
| 22 | 8 August 2014 15:53:00 | 430 | 6 | moderate | FR |
| 23 | 31 August 2015 14:15:00 | 470 | 6 | low | ES, FR |
| 24 | 16 September 2015 11:33:00 | 490 | 5 | low | FR, LU, BE |
| 25 | 13 September 2016 14:57:00 | 630 | 8 | low | FR |
| 26 | 4 July 2018 13:49:00 | 400 | 6 | low | FR |
| 27 | 14 July 2018 20:15:00 | 400 | 7 | low | FR |
| 28 | 7 August 2018 16:50:00 | 420 | 6 | low | FR, BE, NL |
| 29 | 9 August 2018 09:46:00 | 1350 | 18 | low | FR, DE, DK, SE |
| 30 | 12 August 2018 14:29:00 | 710 | 11 | low | ES, FR |
| 31 | 28 August 2018 17:21:00 | 510 | 5 | low | FR |
| 32 | 23 September 2018 12:03:00 | 1220 | 9 | high | FR, BE, LU, DE, CH, AU, CZ, SK |
| 33 | 4 June 2019 16:04:00 | 570 | 6 | low | FR, BE, NL |
| 34 | 5 June 2019 18:06:00 | 510 | 5 | moderate | FR, BE, NL |
| 35 | 16 June 2021 22:06:00 | 414 | 4 | low | FR |
| 36 | 19 June 2021 16:18:00 | 520 | 8 | moderate | FR, BE, LU, DE |
| 37 | 20 June 2021 12:27:00 | 950 | 12 | low | FR, CH, DE |
| 38 | 18 August 2022 01:35:00 | 1400 | 16 | high | ES, FR, IT, SI, AU, CZ |

in Europe from Morel and Senesi (2002). Notably, high activity is observed near mountain ranges such as the Pyrenees, and the Alps, extending roughly to the southern and eastern regions of France, with relatively less activity in Brittany and Normandy, in the northwest. However, the highest frequency of derechos observed in the northeast of France diverges from

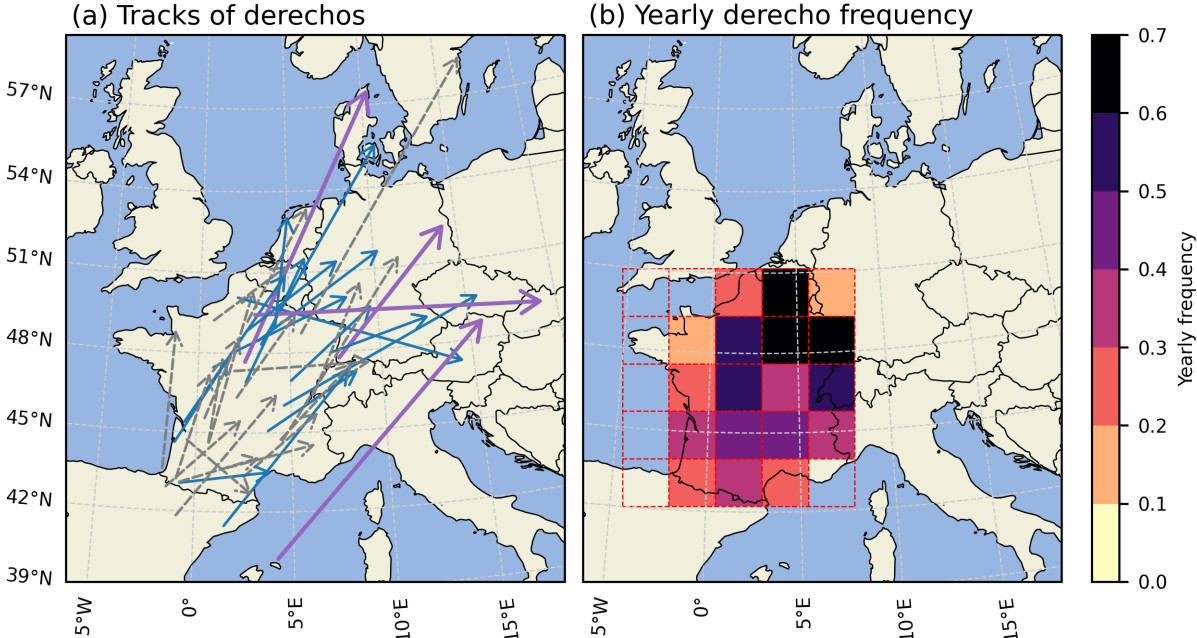

**Figure 1.** (a) Approximate tracks of warm-season derechos that affected France between 2000 and 2022. The tracks are depicted by straight arrows between the first and last severe wind gust reports. The thin broken grey lines, thin blue lines, thick purple lines respectively represent low, moderate and high intensity derechos. (b) Heatmap of the yearly frequency of warm-season derecho computed for geographical cells of dimensions 200 km × 200 km.

these climatologies. Alternatively, when considering the climatology of severe wind events based on reports from ESWD in Taszarek et al. (2020a), we find a better correspondence. Severe wind events are indeed more frequent in central Europe, including Germany, Poland and Czech Republic and to a lesser extent in the northeast of France and the Benelux. Nevertheless, it is essential to emphasize that the reliability of this climatology is constrained by the relatively brief period under consideration (13 years) and the lack of spatial and temporal uniformity in the reports from ESWD. This limitation is particularly pronounced in the case of wind reports, where the frequency has been steadily rising, as documented by Groenemeijer et al. (2017).

In Fig. 2, we depict histograms illustrating the path length, duration, intensity, month of occurrence and start time of observed derechos. The majority of identified events exhibit low (50 %) or moderate (39 %) intensity, with only 11 % classified as high intensity (Fig. 2a). The average path length of derechos in France is 670 km, and a substantial portion of events exhibit a relatively short path length (63 %) (Fig. 2d), falling below the 650 km threshold proposed by Corfidi et al. (2016) for revising derecho definition. Consistent results emerge for event duration, with a majority lasting less than 6 hours (Fig. 2c). The frequency of derechos varies considerably from year to year, reaching peaks of up to 7 events in 2018 and 5 events in 2003, while certain years experience no recorded events (2002, 2008, 2017 and 2020). Successive events also occur, such as the consecutive events on July 26 and 27, 2013, June 4 and 5, 2019, or June 19 and 20, 2021. Transitioning to the analysis of their occurrence by month (Fig. 2b), there is a distinct peak in warm-season derecho occurrences during July, with frequencies

in June and August being comparable, accounting for 84 % of the events combined. The remaining 16 % are distributed across May and September. Examining the start times of derechos (Fig. 2e), we observe that most events initiate in the afternoon, with peaks occurring around 13:00 UTC and again around 16:00 UTC.

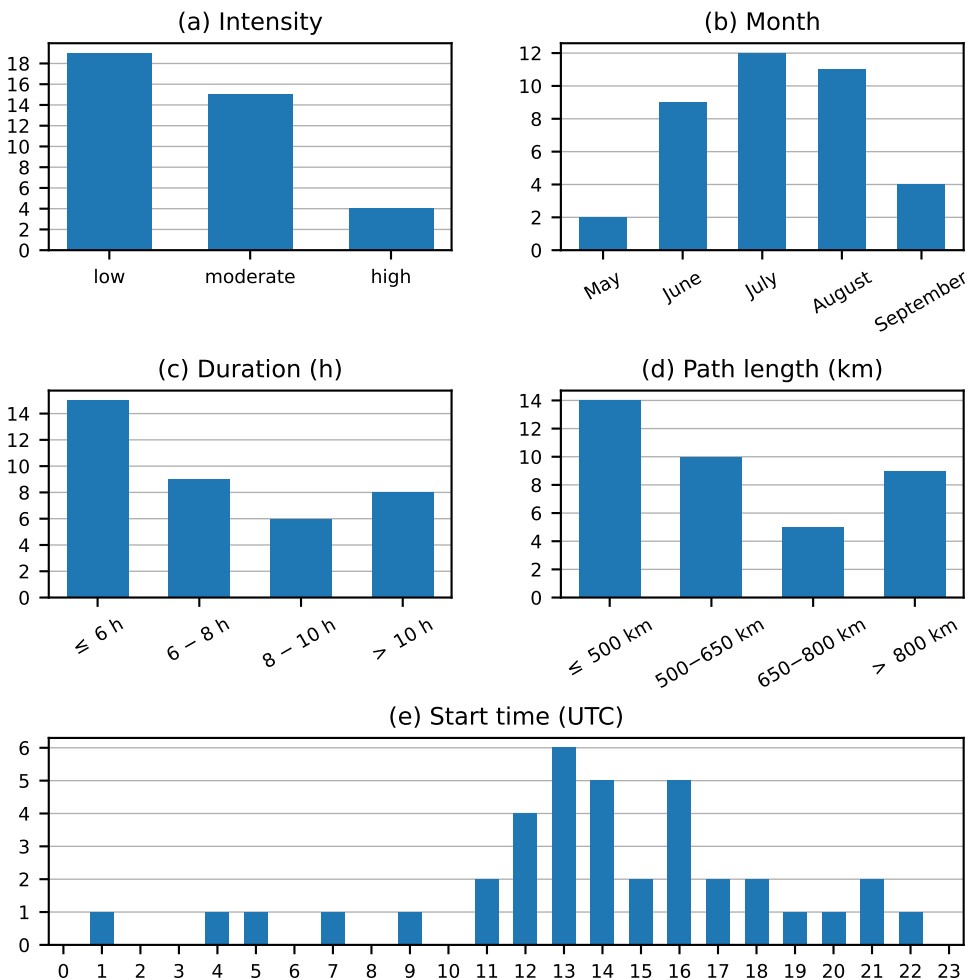

**Figure 2.** Statistics of observed warm-season derechos over France between 2000 and 2022. (a) Intensity defined from the number of reports exceeding given wind gust speed thresholds (high : at least 3 reports $> 38\,\mathrm{m\,s^{-1}}$, moderate : at least 3 reports $> 33\,\mathrm{m\,s^{-1}}$, low : all remaining events). (b) Path length of derechos computed from the distance between first and last severe wind gust reports. (c) Duration defined as the elapsed time between the first and last reports. (d) Month of occurence.

Comparing these results with those from Gatzen et al. (2020) for Germany, we notice a slightly higher frequency of warm-season derechos in France overall (1.65 vs. 1.22 events per year), although it is noteworthy that France is approximately 50 % larger by area than Germany. When adjusting for the size difference, the observed frequency of warm-season derechos in France is slightly lower (1.07 events per year). The highest regional frequency in a standardized 200 km × 200 km square

area in France (0.65 per year) is comparable to the value observed in Germany throughout the entire year, specifically for moderate and high-end events (0.72 per year). In contrast, warm-season derechos are more prevalent in the USA, reaching up to 1.9 events per year for equal-sized grid cells, particularly in the Southern Plains and the Midwest (Coniglio and Stensrud, 2004; Guastini and Bosart, 2016). Regarding trajectory patterns, most warm-season derecho paths in Germany are also directed towards the northeast, aligning with the associated southwesterly flow. In contrast, there appears to be a wider variety of patterns and directions of propagation for warm-season derechos in the USA, with instances of zonal or northwesterly associated circulations (Coniglio et al., 2004). We note smaller proportions of moderate-intensity and high-intensity events (resp. 39 % vs 54 % and 11 % vs 14 %) and a larger fraction of low-intensity events (50 % vs 32 %) in France compared to Germany. However, given the limited sample size, we refrain from asserting the significance of this difference, a conclusion supported by a chi-squared test (Pearson, 1900) at the 0.05 level (p-value = 0.39). These conclusions still hold true when comparing with warm-season derechos in the USA, where the observed proportions are 19 % high-intensity events, 49 % moderate-intensity events, and 33 % low-intensity events (Coniglio and Stensrud, 2004). Peak activity in Germany is also concentrated between June and August, whereas in the USA, it tends to be higher between May and July (Squitieri et al., 2023a). In France, we observe higher activity in August compared to both Germany and the USA. Thus, the distribution seems to lean more towards the late season in France, although this difference lacks statistical significance. If this discrepancy were to be validated with larger sample sizes, it could potentially be attributed to the proximity of the Mediterranean Sea. Its warm waters continue to serve as abundant sources of moisture and heat late in the season, fostering significant instability and the development of severe convective storms (Taszarek et al., 2020a; Morel and Senesi, 2002). Notably, southern France is renowned for experiencing extreme convective rainfall episodes during the fall season (Fumière et al., 2020; Ribes et al., 2019; Taszarek et al., 2019).

## 3.2 Results of attribution for the 18 August 2022 derecho

We initiate our analysis by closely examining the outcomes of the attribution analysis employing the analogues methodology for the 2022 Corsica derecho. This examination aims to illustrate the interpretation of results across various variables and metrics. As introduced in Sect. 1, a MCS developed and advanced northeastward, impacting Corsica, Northern Italy, Slovenia, Austria, and Czechia within a 12-hour timeframe. The event was marked by the generation of robust wind gusts along a 1000 km axis, accompanied by severe hail and substantial rainfall in specific regions. Wind reports from Météo-France (for Corsica) and ISD (mainly for Austria) and the storm's trajectory are depicted in Fig. 3. The synoptic conditions during the storm featured a meridional circulation with a cut-off low situated over the Gulf of Lion in the south of France, as illustrated by the daily average map of Z500 in Fig. 4a. Notably, we observe exceptionally high values of daily maximum CAPE (Fig. 4e) and DLS (Fig. 4f) along the storm's path, particularly over the Mediterranean Sea. These observations underscore the highly favorable conditions that contributed to the initiation of this particularly severe storm. For a comprehensive meteorological report of this event, the reader is referred to ESSL.

Figure 5 presents the results of the attribution study focusing on the synoptic configuration associated with the episode. The Z500 field of the event (Fig. 4a) served as the basis for identifying 37 analogues for both the counterfactual and factual periods, with their composites displayed in Fig. 5a,b. Upon scrutiny, we observe no significant changes in the circulation pattern (Fig.

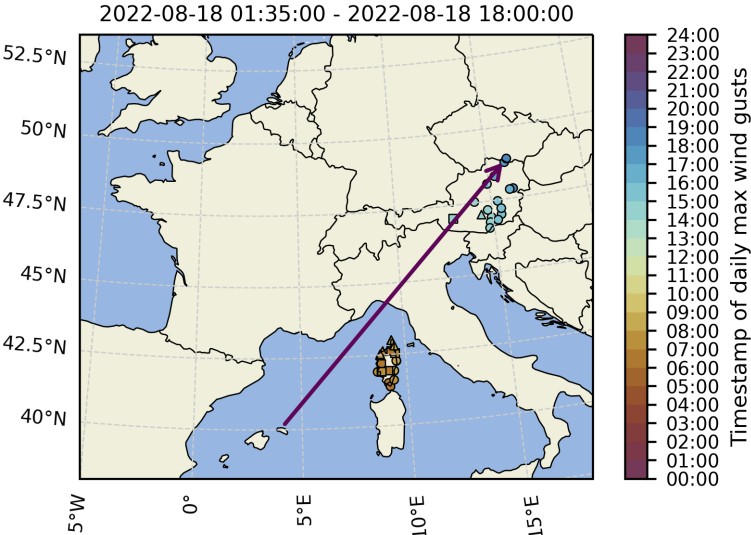

**Figure 3.** Approximate path of 18 August 2022 derecho and locations of severe wind gust reports from Météo-France and ISD colored by their timestamp. The triangles represent extremely severe wind gusts ($> 38\,\mathrm{m\,s^{-1}}$), the rectangles represent medium severe wind gusts ($> 33\,\mathrm{m\,s^{-1}}$) and the circles represent other severe wind gusts ($> 25\,\mathrm{m\,s^{-1}}$). The time stamp of the first and last reports, included those from ESWD are shown in the title.

5c). Notably, there is a pronounced increase in 2 meter temperature across most of Europe, particularly over the Mediterranean Sea. Concurrently, there is a substantial reduction in precipitation over Northern Italy, contrasting with a significant increase in Eastern Europe and certain regions of the Mediterranean Sea (Fig. 5g,h,i). These findings are consistent with results from the E-OBS dataset for temperature and precipitation (Fig. S2 and S3 provided as supplementary materials). Analysis of daily maximum CAPE (Fig. 5m,n,o) reveals a significant surge over the Mediterranean, aligning closely with the exceptionally high values observed on 18 August 2022. It is worth noting that ERA5 values for CAPE exhibits an unrealistic spike for the 2022 derecho (local values exceeding $5000\,\mathrm{J\,kg^{-1}}$ on Fig. 4e), this is a known common issue highlighted in the documentation of ERA5. On the other hand, the examination of deep layer shear (Fig. 5p,q,r) shows no significant signal along the path of the MCS.

The evaluation of analogue quality (Fig. 6a) indicates that the observed circulation is relatively common when compared to the rest of the analogues with no changes between the two periods. Although we do not observe a significant change in persistence $\Theta$ (Fig. 6c) relative to the counterfactual world, there is an apparent increase in the local dimension D (Fig. 6b) in the recent past, signifying a decrease in the predictability of this pattern. Regarding changes in the ENSO, NAO, PDO, EA and SCAND indices (Fig. 6d,e,f,h,i), they are not statistically significant, whereas the distributions of AMO index exhibit a notable shift between the two periods (Fig. 6g), suggesting a potential role of natural variability in explaining the observed changes. A thorough comparison of sea-level pressure, surface temperature, and precipitation patterns characteristic of AMO in Europe (Zampieri et al., 2017) with the changes identified here reveals a remarkable agreement, supporting the notion that

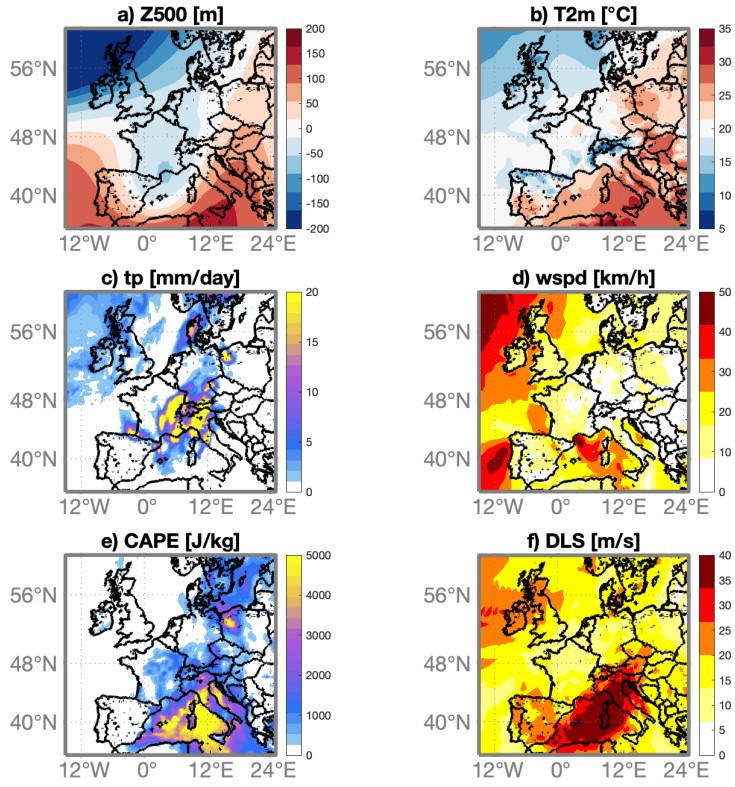

**Figure 4.** Daily averaged maps of geopotential height at 500 hPa, with the spatial mean subtracted (a), 2 meter temperature (b), total precipitation (c), wind speed (d), CAPE (e) and DLS (f) for the 18 August 2022.

the AMO might be a significant influencing factor. The seasonal occurrence of analogues (Fig. 7a) aligns well with the months of lightning and severe wind events in this region (Taszarek et al., 2020a), peaking between August and September, with no significant shift in seasonality observed between the two periods. Lastly, when computing analogues for the entire period and counting their frequency per decade, no discernible trend emerges (Fig. 7b).

In summary, our analysis indicates that there is no significant change in the circulation patterns between the past and present based on the best analogues of the Z500 map associated with the derecho of 18 August 2022. However, it is noteworthy that the surface temperatures associated with the cut-off lows are higher in the current period. A prominent signal of increased CAPE is identified in the present climate, which, with unchanged circulation, can be attributed to the exceptionally high temperature of the Mediterranean Sea. These findings align with the recent research by González-Alemán et al. (2023), highlighting the pivotal role of elevated temperatures in the Mediterranean Sea, characterized by anomalies surpassing +3°C during the summer compared to the seasonal values of 1990–2020, in influencing the development and intensity of the MCS responsible for the 18 August 2022 derecho. However, our analysis underscores the possibility of the Atlantic Multidecadal Oscillation (AMO) contributing to heightened temperatures and CAPE.

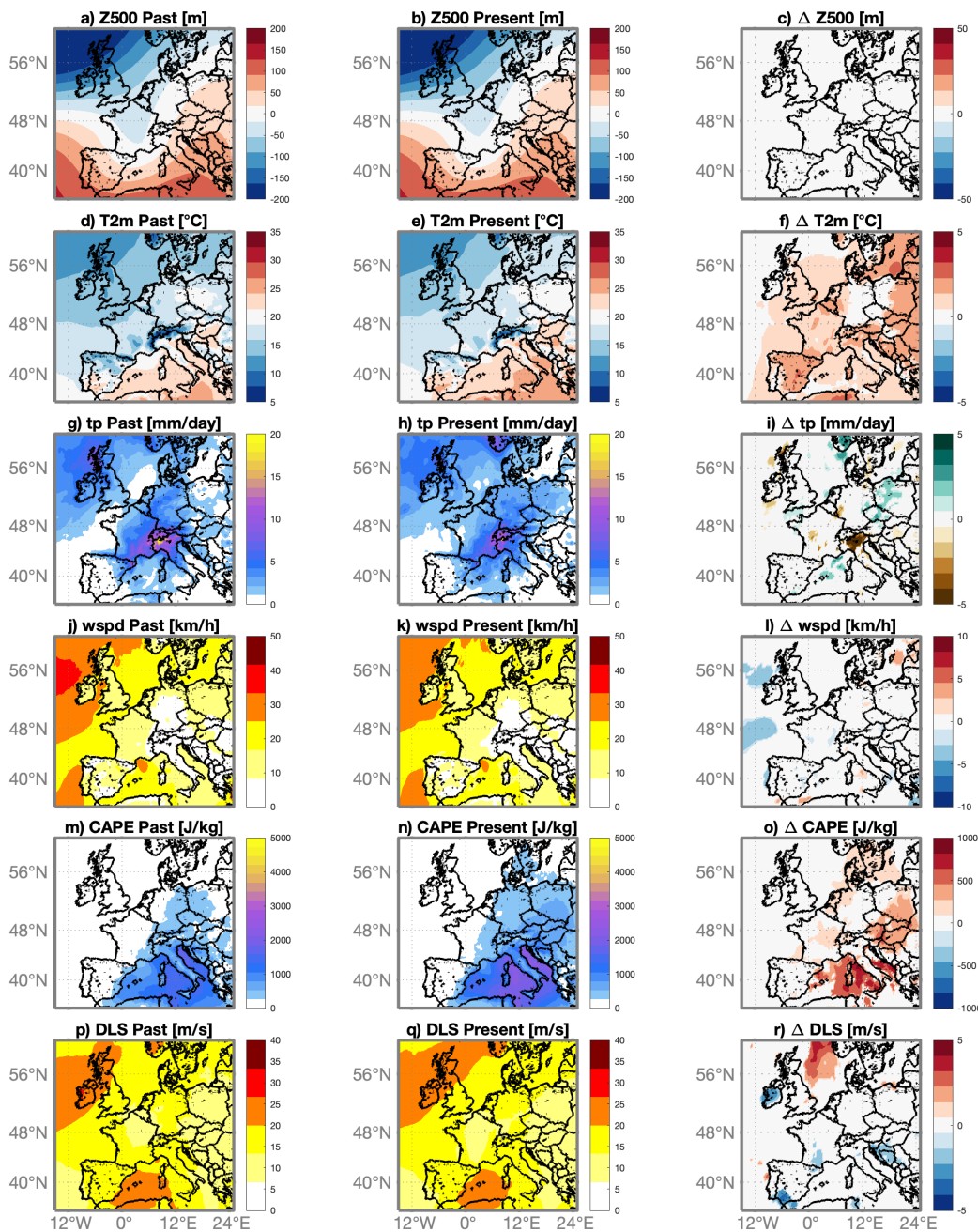

**Figure 5.** Attribution for the 18 August 2022 derecho storm. Average of the 37 analogues of daily mean zero-centered geopotential height anomaly at 500 hPa (Z500) found for the counterfactual [1950–1980] (a) and factual [1992–2022] (b) periods and corresponding 2 meter temperatures (T2m) (d,e), daily total precipitation (tp) (g,h) and wind speed (wspd) (j,k). Changes in the corresponding variables: ΔZ500 (c), ΔT2m (f), Δtp (i) and Δwspd (l) between factual and counterfactual periods (colored-filled areas show significant anomalies with respect to the bootstrap procedure).

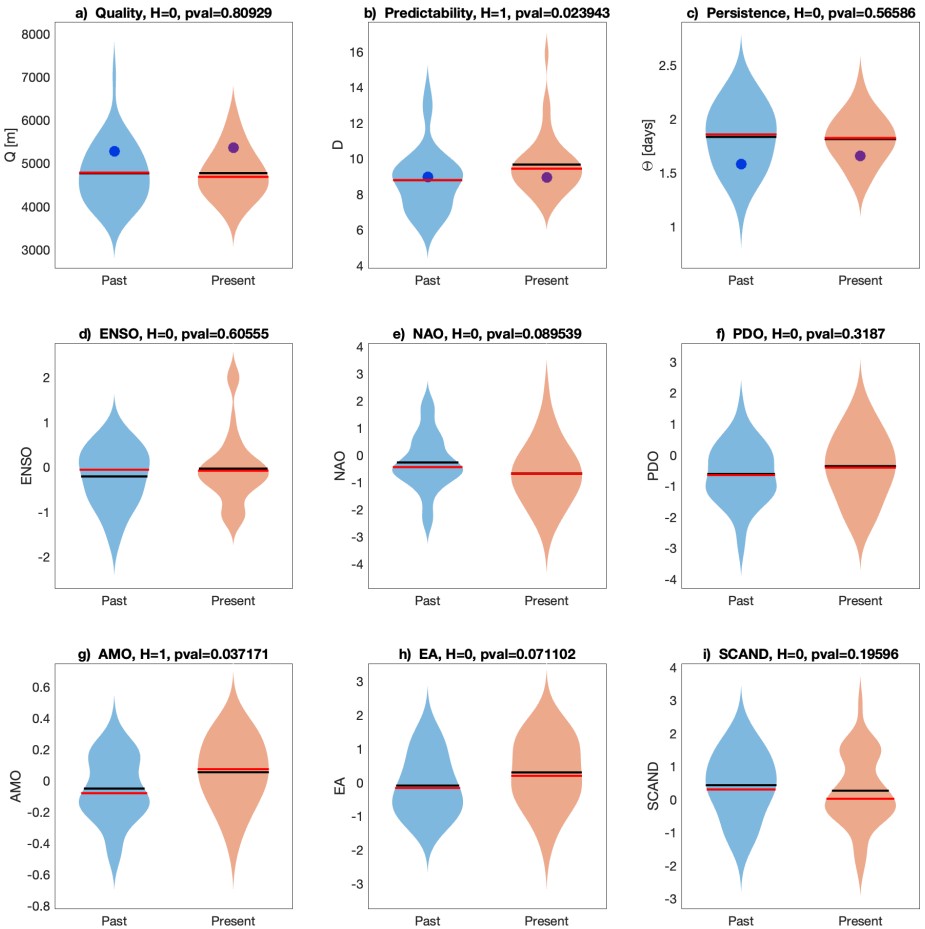

**Figure 6.** Violin plots for counterfactual (blue) and factual (orange) periods for the analogues Quality Q (a) the Predictability index D (b), the Persistence index Θ (c). Violin plots for counterfactual (blue) and factual (orange) periods for ENSO (u), AMO (v) and PDO (w). Values for the day of the event are marked by a dot. Horizontal black and red lines lines respectively represent the empirical mean and median of the distributions. Titles in (a–i) report the results of the Cramér-von Mises test (H) at the 0.05 significance level (H=0 implies that the distributions are compatible and H=1 that they are different) and the p-value (pval).

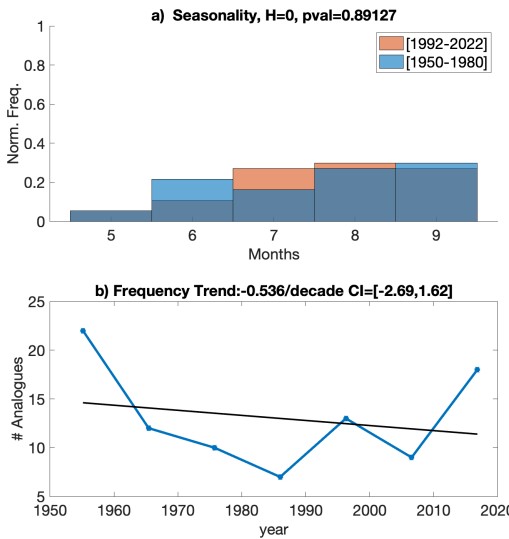

**Figure 7.** (a) Distribution of analogues in each month for the daily mean Z500 map of 18 August 2022. The title report the results of the test (H) at the 0.05 significance level and the p-value (pval). (b) Number of analogues per decade (blue) and its linear trend (black). The title includes the value of the linear trend slope and its confidence interval (CI) in square brackets.

## 3.3 Overall changes in synoptic patterns and environmental proxies

For each event, we conduct a comprehensive analysis similar to the one performed for the 2022 event. The summarized results are presented in Table A1. Subsequently, we go through the overall results for the 38 events.

We identify good-quality analogues for all events, as the event-to-best-analogues distance falls within the distribution of its analogues (not shown). This ensures a meaningful comparison between analogues in the two periods. However, we consistently observe significant changes in the Z500 field (89 %), emphasizing the need for caution when making attribution statements for observed changes in diagnostic variables (T2m, tp, CAPE, and DLS). Some of these changes might result from shifts in circulation patterns, making it challenging to attribute them directly to anthropogenic climate change (Faranda et al., 2023; Vautard et al., 2023). For 42 % of the events, we observe a notable decrease in the average distance (Q) for each analogue compared to its own analogues, indicating that these patterns may be more prevalent in the recent period. However, this does not necessarily result in a substantial positive trend in analogue frequency. Only 5 events (13 %) display a significant positive trend in the frequency of their Z500 analogues, with trend values ranging between 1.4 and 2.3 analogues per decade—a noteworthy change relative to the mean number of analogues per decade. While analyzing the relative frequency of analogue occurrence during the warm season, the majority of results appear statistically insignificant. Nevertheless, in 3 cases, we note a relative increase in frequency during the late season coupled with a decrease in the early season, while an inverse tendency is observed in 3 other cases. We find limited significant results for the local dimension (D), a proxy for predictability, with 3 increasing and 4 decreasing cases. Regarding persistence (Θ), we more frequently observe a significant decrease (24 %), while in 11 % of cases, there is an increase in persistence.

For the majority of events (82 %), we observe a significant increase in temperature, in line with expectations of anthropogenic climate change. There are no cases with a significant decrease, while the remaining cases show no significant change. In almost half of the cases (42 %), we observe an increase in precipitation, while a smaller fraction (18 %) shows a decrease. The increase in precipitation volume aligns with the projected rise in extreme precipitation in Europe (Intergovernmental Panel On Climate Change (IPCC), 2023; Ribes et al., 2019). However, our comparison involves average situations for a given circulation pattern, which may not necessarily correspond to each being an extreme precipitation event. In 74 % of the cases, we observe an increase in instability measured by CAPE, with only one instance of a decrease. This finding aligns with previous studies on the observed or projected rise in convective instability in Europe, particularly in southern Europe (Rädler et al., 2019; Taszarek et al., 2021a, b; Pilguj et al., 2022). Results for DLS show fewer significant trends, with 24 % indicating a decrease and 18 % indicating an increase. These findings are consistent with the more modest and less frequent trends observed in vertical wind shear in the aforementioned studies.

As for the role of natural variability in contributing to these changes, we cannot dismiss its influence on the observed shifts. Examining the factors in descending order of frequency, we find ENSO (74 %), AMO (71 %), SCAND, and PDO (both 37 %), NAO (32 %), and EA (24 %) as the studied variables. These results align with prior studies emphasizing the substantial impact of natural variability on convective activity in Europe (Casanueva et al., 2014; Tippett et al., 2015; Zampieri et al., 2017; Nobre et al., 2017; Piper et al., 2019; Wei et al., 2021). While previous research has suggested a relatively limited influence of ENSO on phenomena such as extreme precipitation in Europe compared to teleconnection patterns (Nobre et al., 2017; Piper et al., 2019), our findings suggest a potentially significant role of ENSO in shaping severe convective environments. Additional investigations are warranted to ascertain the robustness of this observation. One potential explanation for the dominance of AMO and ENSO, among the natural variability factors examined in our analysis, might lie in their longer typical time scales (ranging from annual to multidecadal), in contrast to teleconnection patterns like NAO, EA, and SCAND, which are more closely tied to the chaotic dynamics of the atmosphere and exhibit shorter-term variations on time scale of the order of the week (Hurrell et al., 2003; Hurrell and Deser, 2010). By comparing composites of analogues over 31-year periods, we anticipate that the faster processes would be more effectively smoothed out, emphasizing the influence of longer-term variability.

## 4 Conclusions

In summary, this study presents a 23-year climatology of warm-season derechos in France spanning from 2000 to 2022. To detect events, we used wind gust reports from weather station data, primarily sourced from Météo-France, supplemented by data from ISD, DWD, and ESWD. Mapping was conducted using an MCS detection and tracking algorithm with satellite imagery, leading to the identification and analysis of thirty-eight (38) events. A comparative examination of their features was carried out, drawing parallels with climatologies in the USA and Germany. Derechos in France are notably less frequent than those in the USA and more comparable to those in Germany. Nationwide, an average of 1.65 warm-season derechos occurs per year, with the highest local frequency observed in northeastern France at 0.65 derecho per year within a 200 km × 200 km grid cell. A frequency of occurrence leaning more toward the late season is found in France in comparison with

Germany and the USA, with similar frequencies of derechos in July and August, and a few events in September. Additionally, there is a larger proportion of low-intensity events in France compared to Germany. However, these differences between the two countries cannot be considered statistically significant due to the small number of events. Exploring the potential impact of climate change on altering atmospheric circulation and environmental conditions associated with historical derechos, we compared analogues of circulation patterns based on 500 hPa geopotential height between a relatively distant past (1950–1980) and a more recent period (1992–2022). The analysis revealed a concurrent significant increase in 2 meter temperature and maximum daily CAPE, aligning with findings from other studies in Europe and expectations of a warming climate. However, attributing these changes is challenging due to accompanying shifts in circulation patterns represented by 500 hPa geopotential height. Additionally, natural variability, particularly from ENSO and AMO, cannot be ruled out as contributors to the observed changes.

The methodology employed for derecho detection introduces some limitations, relying on a semi-objective analysis that includes manual decisions, particularly in the selection of days for verifying the existence of associated MCS and mapping wind gust reports with the MCS. While the primary weather station data from Météo-France, DWD, and ISD provides broad coverage, limitations arise in instances where ISD wind gust data is lacking in certain countries or during specific time periods. The scope of this study excludes derechos in the cold season, and a comprehensive full-year climatology remains unexplored. While exploring changes in synoptic conditions and convective environmental parameters, we aimed to estimate the potential impact of specific internal variability factors on large-scale conditions associated with derechos. However, our analysis does not achieve a precise understanding of the individual contributions of these factors and anthropogenic climate change to the occurrence of derecho-producing MCS. Additionally, the study does not consider potential influences from land use and land cover changes, other surface variables like SST, and focuses exclusively on France, without extending its considerations to other regions.

To address these limitations, future studies could benefit from radar data (Huuskonen et al., 2014) and lightning datasets (Schulz et al., 2016) to enhance detection accuracy and refine event definitions. The detection of derechos is arduous due to the collection of data from different sources, the volume of data to analyse and the manual identification procedure. Automation of derecho detection would prevent flaws due to subjectivity and improve efficiency. Additionally, more detailed observations on environmental conditions and dynamics associated with derechos, using proximity soundings (Evans and Doswell, 2001; Gatzen et al., 2020), high-resolution reanalyses such as the forthcoming ERA6 or the analyses provided by non-hydrostatic convection permitting weather models (Coppola et al., 2021), could offer deeper insights. In subsequent studies, a more comprehensive evaluation and quantification of the role of internal variability could be attained by establishing connections between derecho occurrence and indices of these factors. The potential societal impact of derechos on society (Ashley and Mote, 2005), infrastructure, and people's safety, including their correlation with other extreme weather events, would be another interesting aspect to consider. While this study contributes valuable insights, continued research is essential to advance our understanding of the climatology, dynamics, and potential links to internal variability and climate change of derechos in France and Europe.

*Code availability.* The code to compute the dynamical indicators of predictability $D$ and persistence $\Theta$ is available at https://fr.mathworks
.com/matlabcentral/fileexchange/95768-attractor-local-dimension-and-local-persistence-computation.

The Python FLEXible object TRacKeR (PyFLEXTRKR) algorithm developed by Feng et al. (2023a) and is available at https://github.c
om/FlexTRKR/PyFLEXTRKR.

Other analysis codes and the database of warm-season derechosin France are available upon requests from the authors.

*Data availability.* ERA5 reanalysis data is available on the Copernicus Climate Change Service (C3S) Climate Data Store https://cds.cl
imate.copernicus.eu/#!/search?text=ERA5&type=dataset. The ERA5 data for attribution have been downloaded from the preprocessed
http://climexp.knmi.nl. The E-OBS dataset from the EU-FP6 project UERRA (http://www.uerra.eu) and the Copernicus Climate Change
Service, is available from the ECA&D project (https://www.ecad.eu). The GPM IMERG Final Precipitation L3 Half Hourly 0.1 degree
x 0.1 degree V06 and the NCEP/CPC L3 Half Hourly 4km Global (60S - 60N) Merged IR V1 datasets are available from the Goddard
Earth Sciences Data and Information Services Center (GES DISC) respectively at https://doi.org/10.5067/GPM/IMERG/3B-HH/06 and
https://doi.org/10.5067/P4HZB9N27EKU. Weather stations data from Météo-France is available on free request for research purpose from
https://publitheque.meteo.fr/. The Integrated Surface Database (ISD) from the National Oceanic and Atmospheric Administration (NOAA)
is accessible from https://www.ncei.noaa.gov/products/land-based-station/integrated-surface-database. The European Severe Weather
Database (ESWD) from European Severe Storms Laboratory is accessible at https://eswd.eu/. Data from German Weather Service (DWD)
weather stations is accessible at https://cdc.dwd.de/portal).

## Appendix A: Predictability and persistence indices

The attractor of a dynamical system is a geometric object defined in the space hosting all the possible states of the system
(phase-space). Each point $\zeta$ on the attractor can be characterized by two dynamical indicators: the local dimension $D$, which
indicates the number of degrees of freedom active locally around $\zeta$, and the persistence $\Theta$, a measure of the mean residence
time of the system around $\zeta$ (Faranda et al., 2017b). To determine $D$, we exploit recent results from the application of extreme
value theory to Poincaré recurrences in dynamical systems. This approach considers long trajectories of a system—in our case
successions of daily Z500 latitude–longitude maps—corresponding to a sequence of states on the attractor. For a given point $\zeta$
in phase space (e.g., a given Z500 map), we compute the probability that the system returns within a ball of radius $\epsilon$ centered
on the point $\zeta$. The Freitas et al. (2010) theorem, modified by Lucarini et al. (2012), states that logarithmic returns:

$$g(x(t)) = -\log(\text{dist}(x(t), \zeta)) \tag{A1}$$

yield a probability distribution such that:

$$\Pr(z > s(q)) \simeq \exp\left[-\vartheta(\zeta)\left(\frac{z - \mu(\zeta)}{\sigma(\zeta)}\right)\right] \tag{A2}$$

where $z = g(x(t))$ and $s$ is a high threshold associated to a quantile $q$ of the series $g(x(t))$. Requiring that the orbit falls within a ball of radius $\epsilon$ around the point $\zeta$ is equivalent to asking that the series $g(x(t))$ is over the threshold $s$; therefore, the ball radius $\epsilon$ is simply $e^{-s(q)}$. The resulting distribution is the exponential member of the Generalized Pareto Distribution family. The parameters $\mu$ and $\sigma$, namely the location and the scale parameter of the distribution, depend on the point $\zeta$ in phase space. $\mu(\zeta)$ corresponds to the threshold $s(q)$ while the local dimension $D(\zeta)$ can be obtained via the relation $\sigma = 1/D(\zeta)$. This is the metric of predictability introduced in Sect. 2.

When $x(t)$ contains all the variables of the system, the estimation of $D$ based on extreme value theory has a number of advantages over traditional methods (e.g. the box counting algorithm (Liebovitch and Toth, 1989; Sarkar and Chaudhuri, 1994)). First, it does not require to estimate the volume of different sets in scale-space: the selection of $s(q)$ based on the quantile provides a selection of different scales $s$ which depends on the recurrence rate around the point $\zeta$. Moreover, it does not require the a priori selection of the maximum embedding dimension as the observable $g$ is always a univariate time-series.

The persistence of the state $\zeta$ is measured via the extremal index $0 < \vartheta(\zeta) < 1$, an non-dimensional parameter, from which we extract $\Theta(\zeta) = \Delta t / \vartheta(\zeta)$. Here, $\Delta t$ is the timestep of the dataset being analysed. $\Theta(\zeta)$ is therefore the average residence time of trajectories around $\zeta$, namely the metric of persistence introduced in Sect. 2, and it has unit of a time (in this study days). If $\zeta$ is a fixed point of the attractor, then $\Theta(\zeta) = \infty$. For a trajectory that leaves the neighborhood of $\zeta$ at the next time iteration, $\Theta = 1$. To estimate $\vartheta$, we adopt the Süveges estimator (Süveges, 2007). For further details on the the extremal index, see Moloney et al. (2019).

## Appendix B: Detailed results of attribution for all 38 events

In Table A1, we present the detailed results of attribution for all 38 events. Each event's entry indicates whether there is a significant change in the distributions of analogue quality (average Euclidean distance to the best 37 analogues), dynamical indicators (local dimension D and persistence $\Theta$), and indices of natural climate variability (ENSO, NAO, AMO, PDO, EA, SCAND). For analogue quality (Q), local dimension (D), and persistence ($\Theta$), we specify the sign of the change in the mean value using "+" for an increase, "-" for a decrease, and leave the cell blank if there is no significant change. For atmospheric variables or parameters (T2m, tp, using E-OBS, and CAPE and DLS, using ERA5), we apply the same notation. For climate variability indices (ENSO, NAO, AMO, PDO, EA, SCAND), we mark "1" when a significant change in the distribution between the two periods is observed and leave the cell blank otherwise. The same notation is used to indicate significant changes in the Z500 field (from ERA5), which may exhibit complex alterations in synoptic configurations, such as a dipolar or tripolar structure translating into changes in atmospheric flow intensity and/or direction. We also evaluate the significance of potential frequency trends (Trend) and seasonality shifts (S. shift), marking "+" or "-" for positive or negative trends and for shifts towards late or early months of the warm season, respectively.

**Table A1.** Results of the attribution analysis. The columns respectively represent: Number: identification number for each derecho event; Q: analogue quality change: D: local dimension change; Θ: persistence change ; Trend: trend in the number of analogues per decades (+ for increase, - for decrease, blank for no trend); S. shift: seasonality shift in the frequency of analogues (- for significant shift towards the early season, + for shift towards the late season, blank for no shift); Z500 (ERA5): changes in the Z500 pattern (1 if there are significant changes, blank otherwise); T2m (E-OBS), tp (E-OBS), CAPE (ERA5), DLS (ERA5): changes in those variables fields near the MCS path; ENSO, NAO, AMO, PDO, EA, SCAND: changes in the distribution of those natural variability indices between the two periods.

| Number | Q | D | Θ | Trend | S. shift | Z500 | T2m | tp | CAPE | DLS | AMO | EA | ENSO | NAO | PDO | SCAND |
|---|---|---|---|---|---|---|---|---|---|---|---|---|---|---|---|---|
| 1 | - | | | | - | 1 | | + | | | 1 | 1 | 1 | 1 | 1 | |
| 2 | + | | | | | | + | + | + | | | 1 | 1 | | | |
| 3 | | | | | | 1 | + | - | + | | | | 1 | | 1 | 1 |
| 4 | - | - | | | | | + | | | | 1 | | 1 | | | |
| 5 | - | | + | | | 1 | + | + | + | + | 1 | 1 | 1 | | | 1 |
| 6 | | + | | | | 1 | + | + | + | + | | 1 | 1 | 1 | 1 | 1 |
| 7 | - | | | | + | 1 | + | | + | | 1 | | 1 | | | |
| 8 | | | | | | 1 | | | | | 1 | 1 | 1 | | | 1 |
| 9 | - | | | | | 1 | + | + | + | + | 1 | | 1 | | | 1 |
| 10 | | | - | | - | 1 | | + | - | | 1 | | | | | |
| 11 | | + | | | | 1 | + | + | + | | 1 | | 1 | 1 | 1 | 1 |
| 12 | - | | | | | 1 | + | - | + | + | 1 | 1 | 1 | | | |
| 13 | - | | | | + | 1 | + | + | + | | 1 | | | | | |
| 14 | - | | | | | 1 | + | | | - | 1 | | 1 | 1 | | |
| 15 | | + | - | | | 1 | + | | + | | 1 | | 1 | 1 | | 1 |
| 16 | | - | | | | 1 | | + | | | 1 | 1 | 1 | | 1 | |
| 17 | | | | | | 1 | | + | + | | 1 | | 1 | 1 | | |
| 18 | | | | | | 1 | + | - | + | | | | | 1 | 1 | |
| 19 | - | | | | | 1 | + | | + | - | 1 | | | | | 1 |
| 20 | | + | - | | | 1 | + | | | - | 1 | | 1 | | 1 | |
| 21 | - | | + | | | 1 | + | | + | + | 1 | | 1 | | 1 | |
| 22 | - | - | | | | 1 | + | | + | | 1 | | 1 | | 1 | |
| 23 | - | | | | | 1 | | | | - | | | | | | |
| 24 | | | | | | 1 | + | - | | + | 1 | | 1 | | | 1 |
| 25 | - | | | | | 1 | + | + | + | | | | 1 | | | 1 |
| 26 | | | | | + | 1 | + | - | + | | 1 | | 1 | | | 1 |
| 27 | + | + | | | | 1 | + | + | + | - | | | 1 | | 1 | 1 |
| 28 | - | | + | | | 1 | + | + | + | | 1 | | | | | 1 |
| 29 | - | - | + | | | 1 | + | - | + | | | | 1 | 1 | 1 | |
| 30 | - | | | | | 1 | + | | + | - | 1 | | | | | 1 |
| 31 | - | - | | | | | + | | + | - | 1 | | 1 | | 1 | |
| 32 | | - | + | | - | 1 | | + | | + | 1 | | 1 | | 1 | |
| 33 | - | - | + | | | 1 | + | | + | | | | 1 | 1 | 1 | |
| 34 | | - | | | | 1 | + | | + | - | | | | 1 | | |
| 35 | | | | | | 1 | + | | + | - | 1 | | 1 | | | |
| 36 | | - | | | | 1 | + | + | + | | 1 | 1 | 1 | 1 | | |
| 37 | | | | | | 1 | + | + | + | | | 1 | | 1 | | |
| 38 | | + | | | | | + | - | + | | 1 | | | | | |

*Author contributions.* LF performed the detection and tracking of derechos and the subsequent analysis. DF performed the attribution analyses. Both authors contributed to discuss the results and write the manuscript.

*Competing interests.* The authors declare no competing interests.

*Acknowledgements.* We would like to thank the two anonymous reviewers for their insightful and constructive comments, which greatly contributed to improving the quality of this study and refining the manuscript. The authors also thank Bérengère Dubrulle for useful suggestions. The authors acknowledge Météo-France for providing wind gust data from its weather station network. The authors acknowledge the support of the INSU-CNRS-LEFE-MANU (project CROIRE) and ANR-20-CE01-0008-01 (SAMPRACE) grants, from the European Union's Horizon 2020 research and innovation programme under grant agreement No. 101003469 (XAIDA), and the Marie Sklodowska-Curie grant 570 agreement No. 956396 (EDIPI).

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
