# Peer review of "Analyzing 23 years of warm-season derechos in France: a climatology and investigation of synoptic and environmental changes"

_Weather and Climate Dynamics, 2023_

## Author Response (AR1)

We would like to acknowledge the reviewers for their insightful comments and valuable suggestions regarding the scientific quality and relevance of our paper. We have taken the concerns raised by the reviewer seriously and have made significant revisions and amendments to our methodology and manuscript in line with their recommendations. We believe these revisions have significantly enhanced the scientific quality and value of our paper and we hope that our point-by-point answers to each of their major concerns will sufficiently address the issues they have raised.

**I. Reviewer 1**

The paper reconstructs tracks of past major derecho events over France and examines the prevailing environmental conditions in terms of daily mean sea level pressure and near-surface temperature, total precipitation, and wind fields using ERA5 reanalysis. Based on the obtained event set, pressure analogues were estimated as 29-day mean fields for two 30-year periods. Changes in the considered meteorological fields are attributed to climate change. For most of the events studied, the authors found increased precipitation and temperature, while the pressure pattern remained largely unchanged. They also investigated the possible influence of natural climate variability on these changes, based on ENSO and AMO parameters.

I struggle with the scientific quality of the paper and its value to other researchers. My main concerns - among others - are the very small sample of derecho events (see major revision point 1), and that the authors relate the occurrence of derechos solely to near-surface fields. However, while the pressure fields provide the general setting for convection, both MCS and derechos are initiated and maintained by mechanisms in the middle and upper troposphere (e.g., potential vorticity PV anomalies, jet stream-related divergence) and by thermal instability, for which the 2-m temperature is not a reliable proxy (major revision point 2).

See below a list of major and minor revision points and a (short) list of edits.

**Major revision points:**

1. The authors used severe weather reports, mainly from Keraunos, to determine their list of past derecho events. However, in their 30-year study period from 1983 to 2022, they found only 11 derechos, i.e. 37 events per year, but only one in the first 20-year period. I assume that there is either a substantial underreporting of the severe weather reports during the first period, or that the criteria used for the detection failed. For example, Gatzen et al. (2020) identified 40 derechos between 1997 and 2014 that at least partially affected Germany. Therefore, I doubt that the sample is representative for a larger time period. Moreover, I wonder why the authors did not use station data in addition.

We addressed the concern of limited sample of derecho events by detecting past events in France between 2000 and 2022. We decided to focus on warm-season events as one motivation for our study was the derecho of 18 august 2022. We used station data from the French national weather service Météo-France to detect days with concentrated area of severe wind gusts reports and used the PyFLEXTRKR algorithm to automatically detect and track potential associated mesoscale convective system, using satellite imagery (NASA Integrated Multi-satellitE Retrievals for Global Precipitation Measurement (IMERG) dataset). We also used ESWD data to account for reports in other countries. We inspired from the methodology of Gatzen et al (2020) although we didn't use

radar data we think the use of satellite data is a practical approach to study MCS given their global coverage and ease of access, although this induces some limitations that are highlighted in the manuscript. Using this methodology, we significantly increased the sample size to 29 events for the warm-season only.

2. Derechos (or MCS/MCC) are not triggered by surface lows as stated several times (the same applies to T2m). This fact is acknowledged by the authors when they state that "there is no one-to-one correspondence between large scale low pressure systems and the occurrence of derechos" (L60/61). Nevertheless, they quantified composites of analogues based on daily mean sea level pressure. Evans and Doswell (2001), Burke and Schultz (2004), or Coniglio and Stensrud (2004), for example, considered the 500 hPa geopotential height in their investigations of derechos, while Gatzen et al. (2020) additionally considered model-derived PV anomaly and wind shear and stability parameters (e.g., CAPE) from proximity soundings. I strongly recommend considering parameters directly related to strong wind gusts (see, for example, empirical wind gust models, such as those from Wolfson, 1990; McCann, 1994; Nakamura, 1996; Geerts, 2001; or Dotzek und Friedrich, 2009).

Based on the reviewer's suggestions, we have used the 500 hPa geopotential height as a proxy of mid-level atmospheric flow for the analogues search and we included the analysis of changes in convective instability as measured by daily maxima of CAPE and deep layer shear (DLS) defined as 0-6 km wind shear, which are the most common environmental parameters used as predictors for severe convection.

3. The authors rely on ENSO and AMO as proxies for natural climate variability and state that changes in the meteorological fields for episodes negligibly affected by these two modes of climate variability can be attributed to climate change. But what about other low-frequency modes of variability, such as teleconnections (NAO, Scandinavian or East Atlantic patterns) or SST? The reference to Nobre et al. (2017) and their finding that ENSO is important in some regions of the continent is only half the truth. They actually concluded: "that positive and negative phases of the NAO and EA are associated with more (or less) frequent and intense seasonal extreme precipitation over large areas of Europe. The relationship between ENSO and the occurrence and intensity of extreme precipitation in Europe is much smaller than the relationship with NAO or EA, but still significant in some regions".

We agree that other low-frequency modes of variability, such as teleconnections and SST, can also play an important role in extreme weather events. The focus of our study was to investigate the changes in meteorological fields for episodes negligibly affected by ENSO and AMO as these are well-known and widely used climate indices. However, we acknowledge the importance of other modes of variability and agree that they should also be considered in future studies. We have thus included more factors pf natural variability in our analysis (PDO, NAO, EA and SCAND patterns, in addition to ENSO and AMO) as suggested by the reviewer. We included some other references discussing the roles of these differents factor in Europe and revised the part where clarified the interpretation of Nobre et al. (2017) results.

4. ERA5 has too coarse a resolution for reliable estimates of convective gust wind speed and convective precipitation. It is not clear to me why the authors did not use available station data such as E-OBS for their study period.

We have taken into account the reviewer's advice an as mentioned in the answer to point 1., we changed our strategy and used severe wind gusts reports from weather stations to detect and track derechos.

5. Without a detailed interpretation of the results, the discussion of the results is not very scientific. The fact that the 2-m temperature has increased is not a new result, but is stated in every single subsection. I would suggest summarizing the results, especially when they don't really differ (e.g. negligible pressure differences between the two periods found in almost all cases). The use of the same phrases and almost the same wording in subsection 10 is not very clever (e.g., "The analogous analysis is shown in Figure A4").

We appreciate the reviewer's comments and agree that a more detailed interpretation of the results was necessary in the discussion section. We have expanded and improved the analysis of the results and grouped all events (29) to interpret the climatology of detected events (section 2.1) and the results of the attribution study (section 2.3) instead of making separate treatment for each event. We still give a more detailed intepretation for the derecho of August 2022 (section 2.2) to explain how the results shown on Figures 4,5 and 6 are analysed.

6. Why are the derechos tracked using precipitation data? Derechos are defined as contiguous areas of high wind speed, but the relationship to precipitation is not that straightforward.

We have changed our strategy for detecting and tracking derechos as explained in answer to point 1.

7. Introduction: The discussion of how climate change is expected to modify the intensity and probability of convective storms is too general. There are several papers on this that the authors should refer to. The paragraph in the current version implies that there is not much research in this area.

We included a more precise and up to date discussion of the effect of climate change on the intensity and frequency of convective storms, adding more references.

8. Introduction: The objectives of the paper are very vague and imprecise.

We have reworked the introduction to better motivate our study and highlight the purposes of our work, which includes:
- a detection of warm-season derechos in the past 23 years and the analysis of their frequency and characteristics, in comparison with other countries.
- the identification of potential changes in synoptic patterns and environments associated with past warm-season derechos in France, along with a first assessment of the roles of climate change and internal variability.

9. The English writing is (almost) acceptable, but I'd suggest a native speaker / editing service to improve the writing.

We did our best to improve the level of English, taking in consideration the suggestions of the reviewer.

**Minor revision points:**

1. The term "attribution analysis" is misleading when only surface pressure and a few other meteorological parameters are considered.
2. L4: Why writing "in the satellite era" when no satellite data is used or shown? This is somewhat misleading.
3. L8 and elsewhere: What do you mean by "unprecedented"?
4. L11 "clear change in depth of the low pressure system trigger"; more important than the absolute minimum is the pressure gradient; trigger is not the right word either, because a low pressure system does not trigger derechos (otherwise you would have identified a much higher number of derechoes)! The same goes for L13 "low pressure systems possibly leading to derechos".
5. L15: Give a reference when introducing derechos
6. L16: "..serial downburstThis is somewhat misleading because derechos can be divided into serial and progressive derechos.
7. L17: "bow echo": as this is a clear sign of a derecho, briefly explain the term and give a reference.
8. L19-20: "meso-depression"; do you mean a wake low? Again I miss a reference.
9. L28-29 "only a few derechos are registered each year in the world". For example, Bentley and Sparks (2003) identified 230 derechos in a 15-years period in the US; Gatzen et al (2020) detected 40 events in 18 years in Germany. That's pretty much more than a few a year in the world.
10. L30-32: Sentence is unclear; what do you mean by better documented (more damage reports??) "where the available energy is important" is also unclear.
11. L34: "while the large scale conditions" should be specified
12. L40: Is this really true that the entire summer 2022 was governed by a high-low pressure dipole as stated???
13. Figure 1 is too simple. I miss the specification of the geopotential and the altitude, the jet stream, the location of the frontal systems, the warm air advection and so on, see literature. In the caption I do not understand the relation to the polar vortex. Besides, you should refer to the paper by Morris, 1986, who first used the term "Spanish plume".
14. L49-50: Note that the predictability of MCS is much higher compared to isolated convection; can you underpin your statement with a reference?
15. Streamline the data section; in the current version, it's a mix of data and methods and also first results.
16. L85-86: "…consistency of a dataset…" You should mention that over the long term, both the instruments (e.g. radiosondes) and the assimilation of the data (e.g. satellite) have changed, affecting the reanalysis. Therefore, one must be very careful when using them for trend analysis.
17. Method section: Give some more details so the reader can really follow what you did.

18. L112: "The method…" Which method do you refer to? Only the comparison? I wouldn't term this a method.
19. L114-116: This sentence is unclear; "making it statistically impossible to say whether climate change has made" in the context of an unprecedented event is confusing.
20. L118: "Westphalia floods"; the     most severe floods in July 2021 were in Rhineland Palatinate (Ahr basin) and not in Westphalia.
21. "counterfactual world": I wouldn't use this world since the emission of GHG emissions started to increase at the end of the 19th Even if the largest changes in temperature have been observed since the mid-1980s, there were some signs of climate change before that. In this sense, the term "factual world" is not appropriate either.
22. L131: "daily averaged slp": Does this mean you averaged over the 24 hours of ERA5?
23. L140: In what sense did you examine the seasonality?
24. L141: "using monthly indices" of what?
25. L182: "investigate the characteristics of the derechos"; you did not investigate that, but rather prevailing ambient conditions (see also major point 2)
26. L193: Severe hail for sure did not occur along a 1000 km axis.
27. L200: The 2m temperature is not directly related to the convective available potential energy (btw: include the term "available" for CAPE).
28. L204: Give some more details about the orographic influence
29. L250-252: Here the authors refer to mid-tropospheric levels; why not included in the analyses (see major comment 2). Besides, you should write here "(not shown)".
30. L252 and several times later: the authors claim that the derechos originated from an MCC, without giving a reference or explaining how they determined the occurrence of an MCC (according to the extent of different satellite IR temperatures according to Maddox 1980?).
31. L260: "good" is a very subjective term. Can you describe / quantify what you mean by this?
32. L378: I do not agree that you did an in-depth analysis (see major comment 5)
33. L388: Again, low pressure systems do not trigger derechos
34. L408: Why did you mention the Arctic Oscillation at first? Is it really relevant for derechos in France? Do you have a reference for this? What about EA or SCAND?

**Edits:**

1. Authors should use the date (day) consistently; e.g., in L3, they write 18 August 2022, L189/190 August 18, 2022; all subheadings are in the form of dd-mm-yyyy.
2. If you are referring to something you have done once or so, you should use the simple past.
3. L3: of 18 August --> on 18 August
4. L8 reformulate "…that is no good analogues can be found…"; in particular specify "good"
5. L32 reformulate "the public opinion was shocked"
6. L33 "the related MCS formed…"
7. L36: fueled --> maintained or triggered, depending on what you mean; "hot water" for certain is an overstatement (you may specify SST)
8. L38 "there was immediate questioning" reformulate; by whom? Reference? Otherwise this is not really scientific…
9. L43 (and others): I'd prefer the term cut-off low rather than cold drop.
10. L46 use a more scientific formulation for "found guilty"
11. L48 is --> was
12. L50 statements --> analyses / assessments

13. L51 "these modeling difficulties": be more specific
14. L51 "fate" is no appropriate expression
15. L71: The data section should be Section 2 and not 1.1; this affects, of course, all subsequent sections
16. L74: **convective** storm phenomena
17. L75: analysis --> analyses
18. L79: reconstructed
19. L81: "in Figure 2"
20. L81-83: include a reference for ECMWR/ERA5 already here
21. L113: delete "hurricane" as this is completely out of context
22. L117: "attribution protocol" reads somehow weird
23. L126: "natural **climate** variability"
24. L135: prohibit --> limited
25. L101 **an** MCS
26. L196 **Figure** 1
27. L225: of --> in
28. L230 reformulate "unstable context"
29. L265: were --> was
30. 267: "…strong divergence that led to the development of…"; again an MCS
31. L286: Temperature was much warmer…
32. L310 use the abbreviation MCS
33. L323: upper-level minimum pressure (or geopotential height)
34. L367: a minimum **pressure** on

**II. Reviewer 2**

Review of "Changes in synoptic circulations associated with documented derechos over France in the past 70 years" by Fery and Faranda.

In this paper, the authors survey 11 past derecho events that affected France in the satellite era and construct reanalysis-based analogue datasets in order to place these derecho events in the context of two eras with different warming profiles (1950-1979 and 1993-2022). They find two of the events to have no good analogues, but for the events for which analogues can be found, they find that increased precipitation due to higher temperatures is most relevant.

The study puts together a useful dataset of events and analogues, and generally the methods used for the attribution study are sound. My greatest concern is with the emphasis and context of the paper and some of the interpretation (especially early in the paper) that somewhat overpromises based on the relationship between derechos and broad synoptic-scale SLP patterns. With some restructuring, I believe this paper will be a strong addition to the literature.

MAJOR COMMENTS

1. I am finding it difficult to agree with a lot of the analysis presented here mainly because fundamentally low-pressure systems are being used as a proxy for derecho occurrence. While the authors do acknowledge in Lines 60-62 that this is indeed a very weak proxy

relationship, I think some more space in this manuscript should be dedicated to describing the limitations of the use of this low pressure-derecho relationship. What other factors contribute to derecho formation? Might it be possible to see an overall increase in intensity of analogue low-pressure systems but a meaningful decrease in derecho frequency and/or intensity? As I'm sure the authors are aware, one of the substantial difficulties in evaluating the impact of climate change on midlatitude convective storms is that these storms are heavily impacted by mesoscale (and even smaller-scale) processes that are not resolved by the bulk synoptic-scale analogues identified in this manuscript. While I don't disagree that there is very likely a connection between the analogue lows and more intense convective events, I believe this manuscript overstates the certainty in that connection. More time should be spent describing the limitations of equating deeper lows with more derechoes and perhaps also improving and emphasizing the arguments for that connection in a regional sense. I would encourage the authors to reevaluate the organization of this manuscript as a whole, especially in the introduction and conclusions; this is less of a methodological concern and more one of emphasis and context. Fundamentally this is a study of synoptic circulations changing with climate change; the connection to derechos is more tenuous (especially given the very small sample size of 11 events and the lack of comparison to strong low-pressure events that did not result in convection).

We would like to address the major comment regarding the use of low-pressure systems as a proxy for derecho occurrence. To address the concerns raised by the reviewer, we have reformulated our introduction and the presentation of our methodology to clarify our goals and better stress the limitations. Specifically, we have removed the emphasis on low-pressure system and better presented the link between atmospheric circulation patterns and severe convective weather while underlining the importance of sub-synoptic scale phenomena in their development. We also have tempered the interpretation of our results accordingly. We have changed strategy following Reviewer 1's suggestion to consider 500 hPa geopotential height pattern as a proxy of atmospheric circulation and we have included common environmental parameters (CAPE and 0-6 km wind shear) to better investigate changes in the environment and its potential impact on convection.

MINOR COMMENTS

Lines 40-48: While a helpful overview, this paragraph is missing some key citations for the information being presented.

Figure 1: I am a little unclear as to what is being represented by the thermometers to the right of the warm air, since the icons indicate middling temperatures.

Figure 2: The caption for 2i) has the words "tracked between" that don't seem to belong there. In addition, "cumulated" precipitation should be replaced with either "cumulative" or "accumulated".

TRIVIAL/TYPOGRAPHICAL COMMENTS

Line 28: This editorial "as we can see" comment is not necessary and can be removed in place of a more objective description of derecho frequency.

Line 30: The word "the" at the start of the sentence may be omitted. In addition, "Great Plains" is typically capitalized when referring to the region in the United States.

Lines 31-32: "the public opinion was shocked" is an awkward sentence construction. Perhaps "For these reasons, the violence and widespread destruction of the derecho which hit Corsica in summer 2022 garnered a great deal of public attention"?

Line 51: "IPCC reports do not" rather than "does not".

Line 78: "source that documents" rather than "source that document".

Line 87: "this dataset" rather than "this datasets".

Line 89: "reanalysis datasets" rather than "reanalyses datasets".

Line 101: "partially addressed" rather than "partially address".

Line 196: Should read "is given in Table 1".

Line 204: "casts" should be "coasts".

---

## Referee Report (RR1)

**2nd Review „Analyzing 23 years of warm-season derechos in France: a climatology and investigation of synoptic and environmental changes" by L. Fery and D. Faranda (WCD)**

The paper reconstructs tracks of past major derecho events over France and examines prevailing environmental conditions in terms of geopotential height anomalies, 2-m temperature, total precipitation, wind fields, CAPE and shear using ERA5 reanalysis. Based on the obtained event set, analogues in the meteorological fields were estimated as 29-day mean fields for two 30-year periods. The authors attribute changes in the meteorological fields as climate change signals. For most of the events studied, the authors found increased precipitation and temperature, while geopotential height remained largely unchanged. They also examined the possible influence of natural climate variability on these changes, based on ENSO, AMO, and other large-scale teleconnections and parameters.

The authors have done a great job by more or less completely revising the paper. They have taken all of my major revision points into consideration and implemented them in a very satisfying way. This is highly appreciated. In particular, the increase in sample size, the expansion of the event catalog, and the use of convection-related parameters such as geopotential height, CAPE, or vertical wind shear have significantly improved the scientific quality. It also addresses my major concerns about the usefulness of the paper for other researchers.

There are still several issues that need to be addressed before the paper can be accepted for publication. However, these issues can be considered minor revisions. See the list of minor revisions and changes below.

**General:**

1) Why is the time frame limited to May through August? As shown in Fig. 3, July and August have the highest number of derechos, so one can assume that September is also relevant here (in line with the analysis of lightning data from other studies).
2) There are still several flaws in the English writing (see the long list of Edits), and I'd again suggest to consult a native speaker or a proofreading service to improve the writing (e.g. consistent use of tempus, the "s" in the third person singular, use of definite and indefinite articles). In addition, some sentences are very long and complicated, with several subordinate clauses on different topics, which makes reading and understanding very difficult.
3) Once an acronym or abbreviation is defined, it should be used throughout the manuscript.
4) Use NHESS terminology: date is day month year (e.g., 20 July 2020); units must be abbreviated in conjunction with numbers (e.g., speed is 10 km h$^{-1}$ and not 10 km/h).

**Minor revision points:**

1. Condense the first five lines of the abstract as ot is very general. More important is the recapitulation of the main points of the article.
2. L9: Specify what you mean by "changes in synoptic conditions", e.g., by adding "considering two different time periods"
3. L15-17: I'd suggest deleting this sentence in the abstract
4. L68: "typically produces heatwave or stormy conditions"; this is very unspecific and somehow a contraction. Also, "stormy" is not appropriate for convection
5. Whole paragraph L60-75: This is a nice example highlighting the relevance of derechos and the question of climate change. However, I suggest moving this part to the beginning of the

introduction (and shortening this part a bit) before defining derechos and discussing the influences of climate change.

6. L89: Do you have a reference for the stated decrease in relative humidity? Is this also true for CMIP6 data? I know that the older CMIPs show such a decrease in the mid-latitudes. European soundings I looked at a few years ago showed no change in relative humidity.

7. L132-133: "are difficult to disentangle from the associated synoptic-scale winds" This is not really an issue if one simply considers the horizontal pressure gradient.

8. L148: "The reports can come from weather stations"; here you are referring to ESWD, which does not include weather station data.

9. Sect. 2.2: Specify the version of CAPE you used here: mixed-layer CAPE, surface-based CAPE, most unstable CAPE?

   Wind shear is defined as $\frac{\partial v}{\partial z}$, thus the unit is $s^{-1}$; I assume you use wind vector difference as a proxy for shear in this study.

10. L165-166: The sentence "The method ensures that comparisons are relevant .." is unclear

11. L174: A geostrophic flow requires a linear geopotential, which is not the case here.

12. L177: Fronts are not per se sub-synoptic events

13. L184-185: I'm a bit confused here: are the 2-m temperature and daily precipitation totals really from ERA5 data as stated here, or from EOBS as stated in L195?

14. L198: Why 30 years? Both periods are 31 years.

15. L210: Here you may cite Piper et al. "Investigation of the temporal variability of thunderstorms in central and western Europe and the relation to large-scale flow and teleconnection patterns", who investigated the relation between thunderstorm occurrence and NOA, SCAND, and EA patterns.

16. L230: Why do you calculate a linear trend for the entire period when you previously stated that the first period is only marginally affected by climate change?

17. L267: Considering the first and the last report to define the length of a derecho introduces an undefinable level of uncertainty. What about regions and times with significant underreporting? Can you somehow estimate the associated effect? At least a statement about the resulting uncertainty is needed. Also, what is the uncertainty of assuming straight lines for the tracks instead of somehow curved tracks?

18. Tables always have a heading and not a caption.

19. L288-292: What are the reasons for the different hot spots? You can compare this map with lightning data or just speculate. I'd also suggest that identifying four hot spots from a sample of 29 events is not a very robust result. Similarly, how do you explain the diurnal cycle (L310)?

20. Figure 3: The color bar for the time does not make sense as all reports have the same time.

21. Figure 4 and following: I guess it's a problem with Latex that the figures are shown after the literature section?
    I would split Figure 4 into three different figures: one with a-p, one with q-w, and the last with t and x. This would make the discussion much easier.
    In the figure caption, change sea level pressure to geopotential height.

22. The subfigures of Figures 5 and 6 should have the same aspect ratio as that of Figure 4. Also, the captions of Figures 5 and 6 should read "Same as Figures 4a-d, but for...".

23. L306: You should also mention the ESWD reports here.

24. Last paragraph of the conclusions: You may want to phrase new perspectives and outlook in a more affirmative/positive way.

**Edits:**

1. I highly acknowledge the restructuring of the result section and the focus on the main results. However, I doubt whether an appendix of 91 is really helpful. Besides, almost none of the figures in the appendix are discussed in the manuscript, which is not appropriate.
2. Check the brackets in the citations; very frequently \citet{} is used instead of \citep{}; e.g., L42, L113, L116-118, L172, L286, L315-316, L390, L394
3. The terms "high-end" and "low-end" intensity are unclear and not appropriate here. It implies that you refer to both ends of a statistical distribution function rather than to a rough estimation.
4. Be consistent in the use of units; better m s$^{-1}$ instead of m/s
5. L2 there → their; "…and threaten infrastructure"
6. L9 delete "In the second part.." (or otherwise include before "In the first part…")
7. L10: encoded is not an appropriate expression
8. L11: include "…distant past **period**…"; past → period
9. L13: include "**vertical** wind shear" (in contrast to horizontal wind shear)
10. L20: "damaging **winds related to** downbursts" (it's not the downburst with vertical wind component but rather the horizontal wind that produce the damage)
11. L22: "feature predominantly linear characteristics, but also include several bow echoes representing the regions of highest wind speeds" or something like this to resolve the contradiction of "linear characteristics" and "bow echoes"
12. L24: delete "on radar display"
13. L27: "include **a** rear-inflow…"
14. L28: "…wave pattern () oriented embedded within" there's either something mission or you should delete "oriented"
15. L28: "occur**s**"
16. L30: "typically move"
17. L37: reformulate such as "…there must be no more than 3 hours between two consecutive severe wind gust reports…."
18. L39: "emanate"
19. L44: there "…there **are** at least…"
20. L46: derecho**s**
21. L52: "America (USA), …"
22. L57: "In particular, to our knowledge, there is no previous work…"
23. L60: "the public was surprised by"
24. L61: "The associated MCS…"
25. L69 (see also comment 6): "downburst leading to horizontal wind speeds near the surface of up to 225 km h$^{-1}$"; in the original sentence, it reads as if the downburst has that wind speed
26. L72: "…(with anomalies in SST of…"
27. L76: mesoscale convective events → MCS
28. L85: write out CAPE and refer later on CAPE solely
29. L88: delete "midlatitudes and"
30. L89: reformulate "…which makes  any statement about the frequency of severe thunderstorms **difficult**…"
31. L90: "results concerning changes in the …"
32. L93: "and **to** analyze…"
33. L95: deleted "convective available potential energy"
34. L96: "…and **to** introduce…"
35. L98/99: put Section before 2.1 and 2.2
36. L103: use USA only
37. L110: either "a…gust" ore "reporting gusts" if in plural

38. L111: "We then filter out the days that do not have a concentrated area of wind gust reports."
39. L112: to what does "they" refer to?
40. L112: "…there are likely not…" do you mean **they**?
41. L113: include
42. L116: Unclear what you mean by "It"; I think there is a dot missing before Feng et al.?
43. L120: MCSs (Plural)
44. L125: from → since
45. L128: systems
46. L129: "have a global coverage" → "cover large areas"; as you know, single geostationary satellites used for storm detections cover only parts of the globe. So this statement is misleading.
47. L135: mesoscale convective system → MCS; emanate
48. L137: have → has; "… define  a derecho **as** a…"
49. L141: MCSs
50. L145: "…tracks extend" Plural
51. L147: lightning (plural does not exist)
52. L149: doesn't → do not (plural)
53. L150: …"gust speed**. T**his is also a limitation **of our study**,…"
54. L154-155: consider reformulation such as "…the former provide the general setting for convection-favoring conditions in a specific region."
55. L157: "To investigate this" to what refers this?
56. L157: we examined
57. L158: CAPE was already defined
58. L162: climate change started earlier, so "marginally affected" is not appropriate. Besides, "human activity" is too general, better say "greenhouse gas emissions"
59. L164  **annual or multiannual** natural variability"
60. L160-165: a brief discussion about how DLS, CAPE, and Z500 are relevant for convection would be appropriate.
61. L165: "attributed to  climate change ."  (otherwise you should specify the signal)
62. L170: delete "when warming was much more limited" as this occurs several times
63. L175-176: "…which can drive extreme events such as MCS associated with derechos.
64. L178: delete "…and the release of latent CAPE" (btw: it's latent energy or CAPE, but not latent CAPE).
65. L190: large-scale dynamics
66. L198: "We divide the datasets into two periods…" this is now stated for the 4[th] time…
67. L202: delete this sentence
68. L214: unclear what is meant by "the event itself is suppressed."
69. L229: plural p-values and H-test results
70. L247-248: this sentence is a bit cumbersome
71. L260: human activity → climate change (note that "human activity" has a very broad meaning).
72. L275: "…seen **in** Figure**s**…"
73. L278: France have → France has
74. L298: **in** Figure 2
75. L302-303: reformulate "one might obtain upgraded intensities"
76. L307: lower → less
77. L326: an MCS
78. L333: include Figure 4

79. L334: I'd suggest to refer to the geopotential instead of low and high pressure (strictly spoken: your geopotential chart is on the same pressure level of 500 hPa, so there are no high and low pressure systems detectable).
80. L336: "…which  one is the…"
81. Entire Sects. 3.2 and 3.3: correct is "increase/decrease **in**" and not "of" or "on"
82. L338: sea surface temperature → SST
83. L340 for **the** EOBS
84. L342: on 18 August
85. L343-344: "As for deep layer shear 6, we find no significant signal along the path of the MCS". It is unclear to what you refer to as shear was not discussed yet. Besides, to which Figure do you refer?
86. L350 are statistically not significant
87. L351: "() similar to SCAND () suggesting"
88. L373: " **on** average"
89. L376: same as comment 79
90. L380: I do not understand what you mean by "Apart one event for each no good analogue can be found"
91. L388: "… an**d** we observe…"
92. L389: "almost half of the casese…"
93. L400: what is meant by "More some patterns…"?
94. L412: "… **and** the proportion…
95. L414: use plural: MCSs …tend
96. L420-421: New sentence: "Further investigations **are** necessary…and the effect of anthropogenic …"
97. L440: Is that statement necessary? For me it seems to be redundant given the paragraph above.
98. L445: solely SST
99. L449-452: Consider reformulating this sentence as it is very cumbersome.

---

## Author Response (AR2)

We would like to acknowledge again the reviewers for their valuable feedback and constructive criticism on this second version of our manuscript. We have done are best to take into account the suggestions and concerns raised by the reviewers and we have made some substantial modifications to improve the scientific quality of our study. Additionally, we have made conscientious efforts to rectify and refine various aspects of our manuscript. We are confident that these revisions have markedly improved the scientific quality and overall value of our paper. Below are point-by-point answers to the main concerns raised by the reviewers.

**1) Anonymous referee #1 :**

The paper reconstructs tracks of past major derecho events over France and examines prevailing environmental conditions in terms of geopotential height anomalies, 2-m temperature, total precipitation, wind fields, CAPE and shear using ERA5 reanalysis. Based on the obtained event set, analogues in the meteorological fields were estimated as 29-day mean fields for two 30-year periods. The authors attribute changes in the meteorological fields as climate change signals. For most of the events studied, the authors found increased precipitation and temperature, while geopotential height remained largely unchanged. They also examined the possible influence of natural climate variability on these changes, based on ENSO, AMO, and other large-scale teleconnections and parameters.

The authors have done a great job by more or less completely revising the paper. They have taken all of my major revision points into consideration and implemented them in a very satisfying way. This is highly appreciated. In particular, the increase in sample size, the expansion of the event catalog, and the use of convection-related parameters such as geopotential height, CAPE, or vertical wind shear have significantly improved the scientific quality. It also addresses my major concerns about the usefulness of the paper for other researchers.

There are still several issues that need to be addressed before the paper can be accepted for publication. However, these issues can be considered minor revisions. See the list of minor revisions and changes below.

**General:**

1) Why is the time frame limited to May through August? As shown in Fig. 3, July and August have the highest number of derechos, so one can assume that September is also relevant here (in line with the analysis of lightning data from other studies).

We appreciate the reviewer's suggestion to consider the month of September.  We subsequently checked for events in September and found 4 more to include. We also used data from nearby weather stations (from NOAA's Integrated Surface Database (ISD)) in other countries to improve our detection process and the estimation of the intensity of the derechos. After going through the detection process again, we identified more events, bringing the total to 38 (all seasons combined), which is nine more than in the previous version of our study.

2) There are still several flaws in the English writing (see the long list of Edits), and I'd again suggest to consult a native speaker or a proofreading service to improve the writing (e.g. consistent use of tempus, the "s" in the third person singular, use of definite and indefinite articles). In addition, some sentences are very long and complicated, with several subordinate clauses on different topics, which makes reading and understanding very difficult.

We appreciate the reviewer's feedback in identifying grammatical errors and awkward phrasings. We have made dedicated efforts to correct these issues, as well as to rephrase paragraphs and sentences to enhance overall clarity.

3) Once an acronym or abbreviation is defined, it should be used throughout the manuscript.

We thank the reviewer for reporting this issue. We have checked the consistent use of abbreviations.

4) Use NHESS terminology: date is day month year (e.g., 20 July 2020); units must be abbreviated in conjunction with numbers (e.g., speed is 10 km h -1 and not 10 km/h).

We appreciate the reviewer for bringing this to our attention. We have revised the format of dates and units to align with the specified requirements.

**Minor revision points:**

1. Condense the first five lines of the abstract as it is very general. More important is the recapitulation of the main points of the article.

2. L9: Specify what you mean by "changes in synoptic conditions", e.g., by adding "considering two different time periods"

3. L15-17: I'd suggest deleting this sentence in the abstract

4. L68: "typically produces heatwave or stormy conditions"; this is very unspecific and somehow a contraction. Also, "stormy" is not appropriate for convection

5. Whole paragraph L60-75: This is a nice example highlighting the relevance of derechos and the question of climate change. However, I suggest moving this part to the beginning of the introduction (and shortening this part a bit) before defining derechos and discussing the influences of climate change.

6. L89: Do you have a reference for the stated decrease in relative humidity? Is this also true for CMIP6 data? I know that the older CMIPs show such a decrease in the mid-latitudes. European soundings I looked at a few years ago showed no change in relative humidity.

Here are some references that support the claim of a decrease in relative humidity from observations or reanalyses :

Taszarek, M., Allen, J. T., Brooks, H. E., Pilguj, N., & Czernecki, B. (2021). Differing Trends in United States and European Severe Thunderstorm Environments in a Warming Climate. *Bulletin of the American Meteorological Society*, *102*(2), E296–E322. https://doi.org/10.1175/BAMS-D-20-0004.1

Pilguj, N., Taszarek, M., Allen, J. T., & Hoogewind, K. A. (2022). Are trends in convective parameters over the United States and Europe consistent between reanalyses and observations? *Journal of Climate*, 1–52. https://doi.org/10.1175/jcli-d-21-0135.1
However, we do not know about any references supporting these results in CMIP6 models.

7. L132-133: "are difficult to disentangle from the associated synoptic-scale winds" This is not really an issue if one simply considers the horizontal pressure gradient.

While it is accurate that synoptic-scale wind can be approximated using the horizontal pressure gradient, our focus is on local wind gust measurements. These gusts represent fluctuations that may diverge considerably from the average wind speed estimated from the relatively low-resolution synoptic-scale pressure field provided by ERA5.

8. L148: "The reports can come from weather stations"; here you are referring to ESWD, which does not include weather station data.

ESWD actually includes some data from weather stations. The corresponding reports thus feature the recorded wind speed which is useful to assess derecho intensity.

9. Sect. 2.2: Specify the version of CAPE you used here: mixed-layer CAPE, surface-based CAPE, most unstable CAPE? Wind shear is defined as $z$, thus the unit is s -1 ; I assume you use wind vector difference as a proxy for shear in this study.

We thank the reviewer for pointing this lack of precision. We consider the most unstable CAPE from ERA5. We indeed compute vertical wind shear as the wind vector difference between 500 hPa and 10 m. However this metric is usually (improperly) referred to as "0-6 km bulk wind shear" or "deep layer shear". We have clarified these details in the manuscript.

10. L165-166: The sentence "The method ensures that comparisons are relevant .." is unclear

11. L174: A geostrophic flow requires a linear geopotential, which is not the case here.

12. L177: Fronts are not per se sub-synoptic events

13. L184-185: I'm a bit confused here: are the 2-m temperature and daily precipitation totals really from ERA5 data as stated here, or from EOBS as stated in L195?

We use both ERA5 and EOBS for 2-m temperature and daily cumulative rainfall, in order to compare data from reanalysis and observation. In practice the results are almost always compatible but we present only the results of EOBS for these variables in Table 2.

14. L198: Why 30 years? Both periods are 31 years.

We thank the reviewer for reporting this mistake.

15. L210: Here you may cite Piper et al. "Investigation of the temporal variability of thunderstorms in central and western Europe and the relation to large-scale flow and teleconnection patterns", who investigated the relation between thunderstorm occurrence and NOA, SCAND, and EA patterns.

We thank the reviewer for this relevant suggestion, we have included this reference in the text.

16. L230: Why do you calculate a linear trend for the entire period when you previously stated that the first period is only marginally affected by climate change?

In a previous paper, [Faranda, D., Messori, G., Jezequel, A., Vrac, M., & Yiou, P. (2023). Atmospheric circulation compounds anthropogenic warming and impacts of climate extremes in Europe. *Proceedings of the National Academy of Sciences*, *120*(13), e2214525120. https://doi.org/10.1073/pnas.2214525120] we found no significant difference in fitting linear and cubic trends, which motivates this choice in the present study.

17. L267: Considering the first and the last report to define the length of a derecho introduces an undefinable level of uncertainty. What about regions and times with significant underreporting? Can you somehow estimate the associated effect? At least a statement about the resulting uncertainty is needed. Also, what is the uncertainty of assuming straight lines for the tracks instead of somehow curved tracks?

The reviewer has brought up a crucial concern regarding our methodology in defining derecho tracks. In response, we have incorporated a brief discussion addressing the limitations of this methodology in the manuscript.

18. Tables always have a heading and not a caption.

We thank the reviewer for reporting this issue, we put the heading and short caption on top (a short table caption is authorized in the submission guidelines of WCD).

19. L288-292: What are the reasons for the different hot spots? You can compare this map with lightning data or just speculate. I'd also suggest that identifying four hot spots from a sample of 29 events is not a very robust result. Similarly, how do you explain the diurnal cycle (L310)?

It is true that considering the small sample size, it is risky to identify clear hotspots, we have subsequently changed our phrasing to be more descriptive and less assertive in the revised version of the manuscript.

20. Figure 3: The color bar for the time does not make sense as all reports have the same time.

We have included reports from another dataset (ISD, from NOAA) which features weather station data from other countries. Thus we now have reports in Austria and Czech Republic for the 2022 event. Thus we think it makes more sense now to use a colormap for reports'timestamps.

21. Figure 4 and following: I guess it's a problem with Latex that the figures are shown after the literature section? I would split Figure 4 into three different figures: one with a-p, one with q-w, and the last with t and x. This would make the discussion much easier.

We have taken into account the suggestions of both reviewers and split Figure 4 in 4 new figures for better readability. The figure is also displayed at a more relevant place in the main text, which was caused by a Latex issue.

22. In the figure caption, change sea level pressure to geopotential height. The subfigures of Figures 5 and 6 should have the same aspect ratio as that of Figure 4. Also, the captions of Figures 5 and 6 should read "Same as Figures 4a-d, but for...".

We thank the reviewer for reporting these mistakes, we have fixed them.

23. L306: You should also mention the ESWD reports here.

We thank the reviewer for the suggestion, we have now mentionned ESWD.

24. Last paragraph of the conclusions: You may want to phrase new perspectives and outlook in a more affirmative/positive way.

We appreciate the reviewer for bringing this to our attention. We have revised the entire conclusion, including the final section, to present future prospects in a more positive light.

**Edits:**

We thank the reviewer for reporting all these issues. We have done our best to take them into account and improve our manuscript.

1. I highly acknowledge the restructuring of the result section and the focus on the main results. However, I doubt whether an appendix of 91 is really helpful. Besides, almost none of the figures in the appendix are discussed in the manuscript, which is not appropriate.
2. Check the brackets in the citations; very frequently \citet{} is used instead of \citep{}; e.g., L42, L113, L116-118, L172, L286, L315-316, L390, L394
3. The terms "high-end" and "low-end" intensity are unclear and not appropriate here. It implies that you refer to both ends of a statistical distribution function rather than to a rough estimation.
4. Be consistent in the use of units; better m s -1 instead of m/s
5. L2 there ✉ their; "…and threaten infrastructure"
6. L9 delete "In the second part.." (or otherwise include before "In the first part…")
7. L10: encoded is not an appropriate expression
8. L11: include "…distant past period…"; past ✉ period
9. L13: include "vertical wind shear" (in contrast to horizontal wind shear)
10. L20: "damaging winds related to downbursts" (it's not the downburst with vertical wind component but rather the horizontal wind that produce the damage)
11. L22: "feature predominantly linear characteristics, but also include several bow echoes representing the regions of highest wind speeds" or something like this to resolve the contradiction of "linear characteristics" and "bow echoes"
12. L24: delete "on radar display"
13. L27: "include a rear-inflow…"
14. L28: "…wave pattern () oriented embedded within" there's either something mission or you should delete "oriented"
15. L28: "occurs"
16. L30: "typically move"
17. L37: reformulate such as "…there must be no more than 3 hours between two consecutive severe wind gust reports…."
18. L39: "emanates"
19. L44: there "…there are at least…"
20. L46: derechos
21. L52: "America (USA), …"
22. L57: "In particular, to our knowledge, there is no previous work…"
23. L60: "the public was surprised by"
24. L61: "The associated MCS…"

25. L69 (see also comment 6): "downburst leading to horizontal wind speeds near the surface of up to 225 km h -1 "; in the original sentence, it reads as if the downburst has that wind speed
26. L72: "…(with anomalies in SST of…"
27. L76: mesoscale convective events ✉MCS
28. L85: write out CAPE and refer later on CAPE solely
29. L88: delete "midlatitudes and"
30. L89: reformulate "…which makes difficult any statement about the frequency of severe thunderstorms difficult…"
31. L90: "results concerning changes in the …"
32. L93: "and to analyze…"
33. L95: deleted "convective available potential energy"
34. L96: "…and to introduce…"
35. L98/99: put Section before 2.1 and 2.2
36. L103: use USA only
37. L110: either "a…gust" ore "reporting gusts" if in plural38.
38. L111: "We then filter out the days that do not have a concentrated area of wind gust reports."
39. L112: to what does "they" refer to?
40. L112: "…there are likely not…" do you mean they?
41. L113: includes
42. L116: Unclear what you mean by "It"; I think there is a dot missing before Feng et al.?
43. L120: MCSs (Plural)
44. L125: from ✉since
45. L128: systems
46. L129: "have a global coverage" ✉"cover large areas"; as you know, single geostationary satellites used for storm detections cover only parts of the globe. So this statement is misleading.
47. L135: mesoscale convective system ✉MCS; emanates
48. L137: have ✉has; "… define as a derecho as a…"
49. L141: MCSs
50. L145: "…tracks extend" Plural
51. L147: lightning (plural does not exist)
52. L149: doesn't ✉do not (plural)
53. L150: …"gust speed. This is also a limitation of our study,…"
54. L154-155: consider reformulation such as "…the former provide the general setting for convection-favoring conditions in a specific region."
55. L157: "To investigate this" to what refers this?
56. L157: we examined
57. L158: CAPE was already defined
58. L162: climate change started earlier, so "marginally affected" is not appropriate. Besides, "human activity" is too general, better say "greenhouse gas emissions"
59. L164 "long-term annual or multiannual natural variability"
60. L160-165: a brief discussion about how DLS, CAPE, and Z500 are relevant for convection would be appropriate

    We have included a short discussion to justify the use of CAPE, DLS and Z500 in our study.

61. L165: "attributed to the climate change signal." (otherwise you should specify the signal)
62. L170: delete "when warming was much more limited" as this occurs several times
63. L175-176: "…which can drive extreme events such as MCS associated with derechos.

64. L178: delete "…and the release of latent CAPE" (btw: it's latent energy or CAPE, but not latent CAPE).
65. L190: large-scale dynamics
66. L198: "We divide the datasets into two periods…" this is now stated for the 4 th time…
67. L202: delete this sentence
68. L214: unclear what is meant by "the event itself is suppressed."
69. L229: plural p-values and H-test results
70. L247-248: this sentence is a bit cumbersome
71. L260: human activity → climate change (note that "human activity" has a very broad meaning).
72. L275: "…seen in Figures…"
73. L278: France have → France has
74. L298: in Figure 2
75. L302-303: reformulate "one might obtain upgraded intensities"
76. L307: lower → less
77. L326: an MCS
78. L333: include Figure 479.
79. L334: I'd suggest to refer to the geopotential instead of low and high pressure (strictly spoken: your geopotential chart is on the same pressure level of 500 hPa, so there are no high and low pressure systems detectable).
80. L336: "…which is one is the…"
81. Entire Sects. 3.2 and 3.3: correct is "increase/decrease in" and not "of" or "on"
82. L338: sea surface temperature → SST
83. L340 for the EOBS
84. L342: on 18 August
85. L343-344: "As for deep layer shear 6, we find no significant signal along the path of the MCS". It is unclear to what you refer to as shear was not discussed yet. Besides, to which Figure do you refer?
86. L350 are statistically not significant
87. L351: "() similar to SCAND () suggesting"
88. L373: "in on average"
89. L376: same as comment 79
90. L380: I do not understand what you mean by "Apart one event for each no good analogue can be found"
91. L388: "… and we observe…"
92. L389: "almost half of the casese…"
93. L400: what is meant by "More some patterns…"?
94. L412: "… and the proportion…
95. L414: use plural: MCSs …tend
96. L420-421: New sentence: "Further investigations are necessary…and the effect of anthropogenic …"
97. L440: Is that statement necessary? For me it seems to be redundant given the paragraph above.
98. L445: solely SST
99. L449-452: Consider reformulating this sentence as it is very cumbersome.

**2) Anonymous referee #2 :**

In this revision, the authors have changed their focus to 500-hPa geopotential heights and included analysis of CAPE and 0-6 km shear, and they have also increased their attention to factors of internal variability that could be responsible for changes in the environments associated with derecho events. Results that in the previous version were stated as certain are now (appropriately) softened to include more discussion of potential uncertainty.

Unfortunately, while the manuscript is an improvement, in this closer read I do feel as though the scientific quality of this research still feels half-baked; most of my concerns can be summed up by reading through the conclusions and wondering why many of these approaches weren't taken in the first place.

**MAJOR COMMENTS**

1. There are still some major grammatical errors that hinder understanding of the paper. I've tried to highlight a few of the more difficult examples where they come up, as well as some of the minor translation issues. I recommend a more careful read of any future drafts.

We thank the reviewer for pointing out many errors or unclear sentences. We have thoroughly corrected them and checked the quality of our writing.

2. There are several very minor differences that likely disappear in the noise of this relatively small dataset that are nonetheless reported as significant, namely the 1-hr shift in the afternoon peak in French derechos compared to German derechos (Lines 310-311) and the July/August peak in France versus a July peak in Germany and the USA (Lines 315-316).

We thank the reviewer for pointing the limitations due to the small sample size of the dataset. We have conducted statistical tests to assess the significance of the differences between the two datasets and indeed we cannot claim significant differences. Thus, we have more carefully commented these differences and highlighted this issue.

3. There are several instances in the paper where a rigorous result could not be found with the current dataset/methodology and then a reasonable alternative to that approach is brought up immediately, which seriously undercuts the results as presented here. Rather than appearing as future work, these solutions feel like natural steps that should have been taken in the current study. Why is radar/lightning/SYNOP data not included in this analysis? Why are SCAND results not shown when it's one of only two indices showing a significant difference between the times considered?

We recognize that the limitations of our paper have raised concerns. In response, we have expanded our dataset by incorporating weather stations from NOAA's Integrated Surface Database (ISD), which provides synoptic observations, to complement Météo-France, DWD, and ESWD data. However, it's important to note that wind gust data may not be uniformly available in every country, such as Italy, and the time coverage is not always consistent.
In the revised version of our manuscript, we have also considered the month of September to identify potential derechos occurring during this period, given the relatively high activity in August. Through this additional analysis, we have identified 9 more events, including 4 in September, bringing the total to 38 events.
We did not incorporate radar and lightning data mainly due to difficulties in accessing the data and time constraints. We believe our approach is still valuable for an initial exploration of derechos in France, and that it represents a significant contribution to science on severe convective storms in

Europe. Radar and lightning datasets could enhance future studies, refining specific aspects of derechos outlined in our research. We have elaborated on the value of our contribution and provided justifications for our choices in the manuscript.

4. A relatively minor major comment, but whenever p-values are reported, it should be in the context of a predetermined threshold (e.g., significance at the p = 0.05 level) rather than as comparative values for each plot.

We consider changes to be statistically significant when the p-value is less than 0.05, as specified in the text. In the subfigure titles, we not only present the p-value but also indicate the test result as 'H=0' when the null hypothesis, suggesting that the two empirical distributions originate from the same intrinsic distribution, is valid, and 'H=1' when the two distributions differ significantly. We have emphasized this clarification in both the text and the caption of Figure 6.

5. The plots showing comparisons between the 2022 derecho event environment and the mean environments for the two 30-year time periods are not especially helpful and potentially quite misleading, as the smoothing over the 30-year averages will generally always look less "severe" than the environment of the 2022 derecho event.

We appreciate the reviewer's suggestion. In response to input from both reviewers, we have divided Figure 4 into four new figures (Figures 4, 5, 6, and 7) to enhance clarity and readability. Below are answers to some comments.

**MINOR COMMENTS**

Below are answers to some of the minor comments raised by the reviewer.

Line 3: You don't need to focus on the US for a full sentence in your abstract. Consider removing "particularly" onward and add: "Although less frequent than in the USA, derechos also occur in Europe."

Lines 7-8: This feels like introduction material: your abstract should be mostly for summarizing results. "more similar" how? You can summarize all this info in one sentence: "Compared to derechos in Germany, derechos in France are more frequent in August and have a higher proportion of short-lived, relatively low-intensity events."

Line 13: "inconsistent changes" - inconsistent with what?

Line 14: "These changes" - what changes? In derechos? In general?

Line 33: "convective wind-gusts nature" - what does this mean?

Lines 47-48: Briefly explain (in a few words) how they are different

We have added a short explanation about the difference between progressive and serial derechos.

Lines 73-74: How do you justify "very likely" here?

Line 75: Why do you specify convective winds here?

Lines 75-77: I'm not sure I would characterize severe convective storms as "scarce", if that's what you're suggesting here.

Line 88: "midlatitudes and Europe" - these aren't mutually exclusive categories. Is this meant to be midlatitudes in general and Europe in particular?

Lines 100-101: I can't parse the meaning of this description of section 2.2

Lines 111-113: How widespread do you allow the wind reports to be? One of the defining features of a derecho is its substantial horizontal extent (at least along one axis). How do your criteria compare to those of Gatzen et al. (2020)? Why not just use the same criteria?

Indeed we use the same criteria of a long axis of at least 400 km just like Gatzen et al. (2020). We have reworked the Methods section for the detection part to improve clarity.

Lines 115-118: Citations are inconsistent in formatting throughout the paper and should be standardized according to WCD guidelines

We thank the reviewer for reporting this issue. We have fixed the citation formatting.

Lines 118-127: It is unclear to me how this tracking algorithm based on brightness temperature and precip is being incorporated with the wind reports?

We have clarified the use of the tracking algorithm in the revised manuscript. In short, our usage of the algorithm is two-fold :
- detect the existence of an MCS for days when a concentrated are of wind gust reports is found.
- check that the reports match with the detected structure in time and space along a distance greater than 400 km

Lines 136-140: I can't parse the meaning of this sentence.

We acknowledge this sentence was awkward. We have reworked the whole paragraph.

Line 142: Does this mean that you needed at least two reports to be 400 km apart? How does that mesh with Lines 111-112?

In practice, we necessitate a concentrated area, implicitly signifying more than two wind gust reports. We introduced a more precise criterion by imposing the condition that wind gust reports must be within 200 km of each other, with no more than a 3-hour interval between successive reports, following a methodology akin to that of Coniglio and Stensrud (2004).

Lines 148-150: This sentence needs to be revised for clarity.

Line 154: "the former are typical recurrent..." is a poorly constructed sentence and should be revised

Line 164: Should explain here (briefly) how you plan to exclude this low-frequency variability

Line 165: "The method" - which method?

Line 166: How does this method differ from what you consider "statistical modeling techniques"? Some would argue that an analogue approach fits that description.

Lines 167-169: There's a couple conflicting (?) ideas in this sentence that make it difficult to parse.

Line 176: Not sure what is meant by "MCS outbreak" - an MCS may result in severe weather outbreaks, but you wouldn't see an outbreak of MCSs

Lines 192-194: A citation or two for further information on these limitations would be helpful

We added two citations to support these limitations.

Lines 269-270: I assume these categories are mutually exclusive? Should specify; as written, they are not.

Lines 279-281: This statement is unclear and hard to parse

Lines 292-295: This sentence needs to be reworked for clarity

Lines 295-297, 302-303: Why didn't this study include station data from these other countries?

We have incorporated station data from ISD into our analysis. Although this dataset offers international coverage, it may not be as precise and comprehensive as datasets from national weather services (NWS). However, obtaining data from NWS can be challenging and time-consuming due to administrative processes, which were constraints during the revision period. Despite these limitations, the inclusion of ISD enhances the value of our study as an initial exploration of derechos in France. Future studies can build upon our work by utilizing additional datasets to further refine the analysis.

Lines 308-311: Is this difference of 1 hr at all meaningful statistically?

We refer the reviewer to the answer to Major Comment #3.

Lines 315-316: The number of events considered is so small that I don't think you can argue a meaningful difference here.

Figure 3: I cannot interpret this figure at all - what does the colourbar refer to? The arrow is all one colour, as are the icons for reports.

The colorbar refers to the timestamp of the wind gust reports, as explained in the caption. Given the few number of reports, all of them appeared almost in the same color. With the addition of reports from ISD, we now can see reports with different time stamp, as depicted by the different colors from blue to red.

Lines 336-338: This sentence is awkwardly formulated and should be revised for clarity.

Figure 4: The caption is very difficult to follow and should be organized differently

We refer the reviewer to the our response to Major Comment #5.

Line 349: Again, is this shift meaningful enough to note with the small sample size?

Lines 350-352: If SCAND is one of only two statistically significant indices, why is it not shown here?

Lines 358-368: Is "significant" defined here as at the p=0.05 level?

Yes, we refer the reviewer to the answer to Major Comment #4.

Lines 369-378: I think any conclusions drawn here come across as rather speculative and a bit of a stretch given the limitations of the methods and the datasets.

We thank the reviewer for this feedback. We have discarded or reworked some parts that were probably too speculative.

Line 382: The lack of punctuation at the start makes this sentence very difficult to understand

Lines 424-426: If it would have made analysis more rigorous, why didn't you use radar/lightning/SYNOP data?

We refer the reviewer to our answer to Major Comment #3.

**TRIVIAL COMMENTS/TYPOS**

We thank the reviewer for reporting all the issues mentioned below. We have done our best to take them into account and improve our manuscript.

Line 2: "their" rather than "there" (x2)

Line 9: "In a second part" - can be removed

Line 26: "move" should be "propagates"

Line 28: "oriented embedded"?

Line 28: "occurs" rather than "occur"

Line 32: "gust" rather than "gusts"

Line 37: "gust" rather than "gusts"

Line 39: "gust reports must emanate" rather than "gusts reports must emanates"

Line 41: "downburst" rather than "downbursts"

Line 44: "when there are at least" rather than "when there at least"

Line 45: See Line 44 comment

Line 52: No "the" before "derechos"

Line 60: "the public opinion was astonished" - still an awkward construction

Lines 67-68: You say two things ("heatwave or stormy conditions") and then three things ("Eastern Atlantic or Portugal or over Spain") - which corresponds to which?

Line 74: "of the 2022" rather than "of 2022"

Line 78: "much strong" should be "many strong"

Line 79: "we find" should be "the authors find"

Line 87: "MCS tend" rather than "MCS tends" (often we use "MCSs" when pluralizing "MCS", as awkward as that looks!)

Lines 95, 158: You have already defined CAPE and can just use the acronym.

Line 99: "derecho events" rather than "derechos events"

Line 103: No need to define USA here

Line 103: Should give the date again of the 2022 derecho

Line 107: "Derecho detection" rather than "Derechos detection"

Line 110: "wind gust" rather than "wind gusts"

Line 111: "wind gust" rather than "wind gusts"

Line 112: "they are" rather than "there are"

Line 112: "insufficient" rather than "unsufficient"

Line 113: "also include" rather than "also includes"

Line 135: "gusts emanate" rather than "gusts emanates"

Line 136: "gust reports" rather than "gust report"

Line 137: "has some" rather than "have some"

Line 137: "as a swath" rather than "a swath"

Line 141: "gust report" rather than "gusts reports"

Line 145: "extends into" rather than "extends in"

Line 147: "lightning" rather than "lightnings"

Line 170: "height fields at 500 hPa (Z500) to fields from" rather than "height patterns at 500 hPa (Z500) fields to fields from"

Line 171: "that is" rather than "which is"

Line 174: "that is" rather than "which is"

Line 175: "that control environmental conditions that" rather than "that controls environmental conditions which"

Lines 178-179: "latent CAPE" is redundant - just "CAPE" is fine here

Line 184: "2-meter" rather than "2-meters"

Line 191: "this dataset allows us to avoid" rather than "this datasets allows us to avoid"

Line 195: "2-meter" rather than "2-meters"

Lines 204-205: You have already defined ENSO and NAO and can use their acronyms here

Line 223: 3.4 in ENSO refers to the region, not the version

Line 226: "significant" rather than "significance"?

Line 249: Typo: "of the of persistence"

Line 265: "wind gust" rather than "wind gusts"

Line 269: "wind gust reports above thresholds: if there are" rather than "wind gusts reports above thresolds: if there"

Line 278: "each cell" rather than "each cells"

Line 278: "the northeast of France has the highest" rather than "northeast of France have the highest"

Line 279: "no events" rather than "no event"

Lines 281-283: Don't need to start this sentence with "However"; "1.9 events per year for equal-sized grid cells" rather than "1.9 event per year for equal size grid cell"

Line 288: "and propagates" rather than "and move"

Line 298: "in Figure" rather than "on Figure"

Line 301: "and a larger" rather than "and larger"

Line 302: "weather station" rather than "weather stations"

Line 307: Seem to be missing words between "results" and "that"

Line 310: "late afternoon" rather than "end in the afternoon"

Line 319: "Germany or" rather than "Germany of"

Figure 2: In caption, "wind gusts speed" should be "wind gust speed"

Figure 3: In caption, "rectangle" should be "rectangles"

Line 336: Typo: "which is one is"

Line 342: "observed on" rather than "observed in"

Line 343: The "6" can be removed here.

Line 371: "translate to" rather than "translates in"

Line 373: "on average" rather than "in average"

Line 376: Word missing after "southwesterly"

Line 387: "a significant" rather than "an significant"

Line 398: comma splice and "an" instead of "and" - there are many minor errors throughout the rest of the paragraph and section

Line 406: "wind gust" rather than "wind gusts"

Line 412: missing words

Line 414: "tend to" rather than "tends to"

Line 424: "gust reports" rather than "gusts reports"

Line 439: I don't think "dispose of" is the phrase you want here

---

## Author Response (AR3)

**Co-editor decision: Publish subject to minor revisions (review by editor)**
by Peter Knippertz

The authors have worked very hard to improve the manuscript. It is good to see that the time period was extended and that more stations were used. Also the language appears to have improved overall. Although not all points of the reviewers have been fixed to 100% satisfaction, I exchanged with the editor in chief, Heini Wernli, and we decided to accept the paper now subject to minor revision. Nevertheless I would like to call on the authors that next time they should not submit manuscripts in a somewhat pre-mature state. This takes too much time and energy of reviewers and editors, who give all this for free.

The technical fixes that we would like to see are listed below. In addition, I would like to ask you once again to go through the manuscript and eliminate remaining unclear formulations or other errors you may find. Ideally, you do this with the help of an outside native speaker.

**Author's response:**
We sincerely appreciate the efforts of the editor and reviewers in carefully evaluating our manuscript and providing valuable feedback. We are pleased to see that the extension of the time period and the inclusion of additional stations have been recognized, along with improvements in the language throughout the manuscript. While we acknowledge that not all reviewer suggestions have been addressed to full satisfaction, we are grateful for the decision to accept the paper pending minor revisions.

Moving forward, we understand the importance of submitting manuscripts in a more polished state to minimize the burden on reviewers and editors. We will take this feedback into consideration for future submissions.

In response to the technical fixes listed by the editor, we have carefully reviewed the manuscript and made necessary adjustments.

We thank the editor and reviewers once again for their patience and understanding throughout this revision process.

Below, we answer to some of the minor revision points raised by the editor.

line 5: "Similar to Germany" is strange, given that the previous sentence says that you look at France. I think this happened because you often describe in your paper your results for France with results from the literature for Germany and the US. But this must be much clearer throughout the paper. Here in the abstract, I would suggest that you first describe your results for France, and then add a remark about how things compare with results for other countries that were previously published.

We have reworked slightly the abstract to put more emphasis on the results for France, as suggested by the editor.

line 6: what is "a suggestive trend"? Is that a trend that is not statistically significant? I think the abstract should focus on results that are solid, and not on things that should be taken with caution.

We value the editor's feedback, and in response, we have removed references to uncertain outcomes from the abstract.

line 11: "consistent with Mediterranean trends" is not clear to me, are these trends from other regions / other studies?

> We have clarified this in the abstract (we meant previous studies).

line 60: no need for "In 2016"

line 97: space after "hazards" is missing

line 120: sentence is very verbose, maybe just "Finally, conclusions and future perspectives are presented in Sect. 4.

line 146: something wrong with formatting of reference; same on line 147 and 148

line 156: inconsistent use of tense: one sentence is past tense, the next present tense …

line 161: "… days when" (not where)

line 176: "what is an initial climatology"? the same as a preliminary climatology?

> We changed "intial" for "first".

line 185: most likely "." missing?

line 186: I don't understand what is meant by geopotential height offset

> We have clarified this in the revised manuscript.

line 190: I don't understand this sentence, how can the same analogs capture both heat waves and cold waves? Why do you need to mention here cold waves?

> We have reworked this part, also removing mentions of many types of extreme events as suggested in the next revision point.

line 207: what is meant by "leading up to"? Not sure that you need to list here all previously studied events

line 253: I don't understand why you mention influences of ENSO etc. on "some regions in Europe", when this study is specifically about France

> We have fixed this sentence.

line 282: I don't understand, I thought the analogs have a small Eucledian distance per definition, so how can the event not "belong to the same distribution"?

> We have clarified this point in the revised manuscript.

line 302: I think you explained this already in Sect. 2?? no need for repetition.

> We have removed the details on this classification that were presented in the introduction and in section 3.1 and moved them to section 2.1 instead.

line 305: I am irritated that you start presenting your results with "In comparison to Germany". This sounds to me as if you studied with the same data derechos in Germany and France and now make a comparison. But this is not the case(?). I would present your results, and then in a separate paragraph explain whether your results differ or not from results published by others for Germany and the US and … I would restructure this entire Section.

> We thank the editor for pointing out this awkwardness. We have entirely reworked this section as suggested by the editor.

line 313: please check again that you don't repeat all methodologies details here that you already introduced in Sect. 2.

> We have checked that no repetition was made between Sect. 2. and 3.

line 379: don't include "figure caption explanations" in the main text of the paper.

everywhere "Figure" should read "Fig." (except for beginning of sentences).

Fig. 4: explain what a zero-centred anomaly is

> We have explained that the spatial mean has been removed from the Z500 field in the caption.

Fig. 5: panels are very small, no need to have longitude labels below all panels, in this way panels could be enlarged.

> We have removed the longitude labels between the panels, and we have enlarged the panels.

Fig. 7: it seems to me that with so few events not statistical test is required, these differences / trends can hardly be significant. I would just show the data, mention that there is not enough data for a statistical analysis and not burden the reader with H=0 and pval=0.89127 …

> We believe that showing the results of these statistical tests is relevant as for some events, the tests are found to be negative (i.e. there are significant differences in the seasonality or significant trends, as shown in Table 2). Consequently, we chose to let the results of these tests in the titles of Figure 7, as in Figure 6.

What should the reader get from Table 2? Maybe this should go to a supplement … I have the impression that this analysis can be shortened, also because you very often mention "the need for caution" and because is written in a way that leads the reader a bit puzzled about what you try to convey.

> In accordance with the editor's recommendation to condense Section 3.3, Table 2 has been relocated to Appendix B, along with its corresponding descriptive paragraph.

line 462: careful when comparing your results for a few events with general statements about convection!

> We have added a sentence to highlight the need for further investigation to check the robustness of these results.